# Atomically dispersed golds on degradable zero-valent copper nanocubes augment oxygen driven Fenton-like reaction for effective orthotopic tumor therapy

Liu-Chun Wang [1], Li-Chan Chang [2], Wen-Qi Chen[1], Yi-Hsin Chien [3], Po-Ya Chang [4], Chih-Wen Pao[4], Yin-Fen Liu[2], Hwo-Shuenn Sheu [4] ✉, Wen-Pin Su [2,5] ✉, Chen-Hao Yeh [3] ✉ & Chen-Sheng Yeh [1] ✉

Herein, we employ a galvanic replacement approach to create atomically dispersed Au on degradable zero-valent Cu nanocubes for tumor treatments on female mice. Controlling the addition of precursor $HAuCl_4$ allows for the fabrication of different atomic ratios of $Au_xCu_y$. X-ray absorption near edge spectra indicates that Au and Cu are the predominant oxidation states of zero valence. This suggests that the charges of Au and Cu remain unchanged after galvanic replacement. Specifically, $Au_{0.02}Cu_{0.98}$ composition reveals the enhanced •OH generation following $O_2 \rightarrow H_2O_2 \rightarrow$ •OH. The degradable $Au_{0.02}Cu_{0.98}$ released $Cu^+$ and $Cu^{2+}$ resulting in oxygen reduction and Fenton-like reactions. Simulation studies indicate that Au single atoms boot zero-valent copper to reveal the catalytic capability of $Au_{0.02}Cu_{0.98}$ for $O_2 \rightarrow H_2O_2 \rightarrow$ •OH as well. Instead of using endogenous $H_2O_2$, $H_2O_2$ can be sourced from the $O_2$ in the air through the use of nanocubes. Notably, the $Au_{0.02}Cu_{0.98}$ structure is degradable and renal-clearable.

The successful preparations from the groups of Flytzani-Stephanopoulos and Zhang have given the unforeseen demonstration on single-atom catalysts (SACs) and have brought nanocatalysis into atomic age[1,2]. SACs possess atomically dispersed structures and well-defined coordination architectures, making them useful in both heterogeneous and homogeneous catalysis[3–5]. Taking advantage of the endogenous $H_2O_2$ in the tumoral microenvironment, SACs have emerged as a form of nanocatalytic medicine used to perform chemodynamic therapy following •OH generation. In addition to the well-known Fenton or Fenton-like actions, SACs also show enzyme mimicking activity, e.g. a peroxidase (POD)-like reaction, to produce •OH in tumor catalytic therapy. That is, these SACs are viewed as nanozymes, revealing catalytic activity with enzyme-like properties due to their unique electronic structures and their specific coordination environments to catalyze versatile reactions[6–8]. Undoubtedly, SACs promote nanozymes at the atomic level and thus provide potential workarounds for current inherent limitations.

Previous studies typically used Fe single-atom or peroxidase mimicking catalytic activity to generate •OH for single-atom-based tumor catalytic therapy. Importantly, the studies all used $H_2O_2$ from the tumor microenvironment (endogenous $H_2O_2$) to perform tumor catalytic therapy. However, because of the cellular redox homeostasis, the intratumoral $H_2O_2$ level is usually below the threshold (ca.100 μM), which limits the therapeutic effect of •OH. Thus, alternative

[1]Department of Chemistry, National Cheng Kung University, Tainan 701, Taiwan. [2]Institute of Clinical Medicine, College of Medicine, National Cheng Kung University, Tainan 704, Taiwan. [3]Department of Materials Science and Engineering, Feng Chia University, Taichung 40724, Taiwan. [4]National Synchrotron Radiation Research Center, Hsinchu 30076, Taiwan. [5]Departments of Oncology and Internal Medicine, National Cheng Kung University Hospital, College of Medicine, National Cheng Kung University, Tainan 704, Taiwan. ✉e-mail: hsheu@nsrrc.org.tw; wpsu@mail.ncku.edu.tw; chenhyeh@fcu.edu.tw; csyeh@mail.ncku.edu.tw

approaches have been employed to boost the availability of reactive oxygen species (ROS) for tumor catalytic treatments. For example, parallel catalytic behavior was found to simultaneously generate superoxide ion ($O_2\bullet^-$) and •OH following peroxidase-like catalytic activity[9]. A co-catalysis approach was designed by taking $MoS_2$ support as a co-catalyst to accelerate the conversion of $Fe^{3+}$ to $Fe^{2+}$ for Fenton reactions[10]. Integrated cascade reactions have also been proposed based on Fenton- and peroxidase-like activities to concurrently generate •OH and $O_2\bullet^-$[11]. Recently, a MOF-derived flower-like structure was created to provide 3D accessibility of active sites to boost •OH following peroxidase-like activity[12]. Apart from the use of endogenous $H_2O_2$, recently copper hexacyanoferrate forming the single-site nanozyme was fabricated to process glutathione-oxidase, resulting in $H_2O_2$ formation accompanying the conversion of $Cu^{2+}$ to $Cu^+$ for the subsequent Fenton-like reaction leading to •OH[13]. In this work, the single-atom Au on the zero-valent Cu nanocubes was fabricated to effectively reduce $O_2$ to form $H_2O_2$ for the following •OH production (Fig. 1a).

A general approach to creating atomically dispersed metal typically uses a heterogeneous active metal as a single-atom anchored on support following bottom-up methods, such as adsorption following impregnation and co-precipitation, or top-down methods using pyrolysis to conduct evaporation process for the generation of single-atom sites[14,15]. On the other hand, galvanic replacement is a process that follows a redox reaction to gradually replace one metal with another metal ion with a higher reduction potential to construct alloyed nanostructures, thereby creating diverse morphologies, shapes, and sizes. Galvanic replacement can thus be viewed as a hybrid bottom-up and top-down approach. This reaction process has great potential to achieve single-atom site formation by controlling either the number of metal ions added (higher reduction potential) or the reaction period of galvanic replacement. However, few reports have described the use of galvanic replacement to manipulate SACs and are largely limited to the catalysis of hydrogenation or glucose oxidation from Cu/Pt$_{(SA)}$, Cu/Pd$_{(SA)}$, Ni/Pt$_{(SA)}$, and Pd/Au$_{(SA)}$[16–21]. Here, we use a galvanic replacement reaction to fabricate the Au single atoms on zero-valent copper nanocubes. The reduction potentials of copper (+0.522 V, $Cu^+/Cu$; +0.341 V, $Cu^{2+}/Cu$) are more negative than that (+0.695 V) of $O_2/H_2O_2$, making zero-valent Cu thermodynamically feasible to reduce oxygen to $H_2O_2$. Therefore, the zero-valent Cu can be an $O_2$ activator for •OH production. Specifically, the resulting $Au_{0.02}Cu_{0.98}$ composition reveals the enhanced •OH generation, making it a promising chemodynamic agent, following $O_2 \rightarrow H_2O_2 \rightarrow •OH$. The degradable $Au_{0.02}Cu_{0.98}$ released $Cu^+$ and $Cu^{2+}$ resulting in oxygen reduction and Fenton-like reactions. Instead of using endogenous $H_2O_2$, $H_2O_2$ can be supplied from the $O_2$ under aerobic conditions. Simulation has indicated that the Au single atom facilitates zero-valent copper to reveal the catalytic property of $Au_{0.02}Cu_{0.98}$ composition for $O_2 \rightarrow H_2O_2 \rightarrow •OH$ as well. Furthermore, the $Au_{0.02}Cu_{0.98}$ structure is degradable under acidic conditions, which favors the excretory system through urinary metabolism when applied in tumoral treatments.

## Results and discussion

### Characterization of Au/Cu⁰ nanocubes

Hydrophobic-based Cu nanocubes were prepared and implemented by oil/water phase transformation with cetyltrimethylammonium bromide (CTAB) surfactant and polyvinylpyrrolidone (PVP) treatments in a water phase. The characteristic results of Cu nanocubes are shown in detail in Fig. 1b, c and Supplementary Fig. 1. The Cu nanocubes have a surface plasmonic resonance (SPR) absorption band at 580 nm (Fig. 1b) and an edge length of 50 nm (Fig. 1c and Supplementary Fig. 1a). The XRD results and electron diffraction pattern show a well-defined crystalline structure for the Cu nanocube corresponding to a face-centered cubic (fcc) crystal structure with (111), (200) and (220) faces (Supplementary Fig. 1b–e). The various ratios of Au/Cu⁰

nanocubes were synthesized through a galvanic replacement reaction by tuning the addition of precursor HAuCl$_4^-$. The oxidized Cu atoms were replaced by Au⁰ atoms owing to the different redox potential between $Cu^{2+}/Cu$ (0.34 V vs. the standard hydrogen electrode [SHE]) and $AuCl_4^-/Au$ (0.99 V vs. SHE). The SPR bands of Au/Cu⁰ nanocubes gradually red-shifted to 590 nm following the increase of Au atoms, $Au_{0.02}Cu_{0.98} \rightarrow Au_{0.05}Cu_{0.95} \rightarrow Au_{0.1}Cu_{0.9} \rightarrow Au_{0.5}Cu_{0.5}$ (Fig. 1b). The fixed concentration of Cu nanocubes ($10^4$ ppm, 100 µL) was used for galvanic replacement reaction accompanied with 50 mM HAuCl$_4$ in the volumes of 5, 10, 20, 40, 60, 80, and 100 µL, respectively. A 5 µL of HAuCl$_4$ obtained $Au_{0.02}Cu_{0.98}$. The ratios of different compositions for nanocubes are determined by AA measurements. The corresponding TEM images of the nanocubes can be seen in Fig. 1d–g. The morphology change could be observed beginning from solid to the rough surface and then turning into a cage-like structure as the Au atoms increased. For comparison, Au nanocubes with similar edge lengths[22] were also fabricated for further studies (Supplementary Fig. 2). Au nanocubes have an SPR band at 535 nm (Fig. 1b). It is noted that the composition and morphology of $Au_xCu_y$ could be further transferred into alloyed nanocages ($Au_{0.75}Cu_{0.25}$), nanoframes ($Au_{0.8}Cu_{0.2}$), and the completed replacement yielding Au nanoparticles with the addition of the volumes of HAuCl$_4$ in 60, 80, and 100 µL, respectively. The characteristic results including the TEM images, XRD patterns, and HR-TEM images were shown in Supplementary Fig. 3 to Fig. 5. The alloyed nanocages ($Au_{0.75}Cu_{0.25}$) and nanoframes ($Au_{0.8}Cu_{0.2}$) could be obviously observed by TEM images and the corresponding XRD patterns were assigned to fcc crystal structure individually followed JCPDS cards 01-071-5023 and JCPDS cards 01-072-5241. In addition, the HR-TEM images and their related electron diffractions indicate the arrangements of (200) and (110).

Synchrotron powder X-ray diffraction (XRD) measurements were performed to verify the face-centered cubic (FCC) crystal structure of the Au/Cu⁰ nanocubes. Details of the material characterization experiments are given in the Supporting Information. As shown in Fig. 1h, the characteristic peaks in the XRD patterns of Au/Cu⁰ nanocubes can be attributed to Au (COD ID. 1100138) and Cu (COD ID. 4105040) in the Fm-3m cubic phase. The sharp peaks for Cu nanoparticles indicate larger crystal sizes, while the broadened peaks are indexed to gold-rich crystalline structures with smaller grain sizes. The grain size was calculated based on the peak width according to the Scherrer equation (Supplementary Table 1). The grain sizes of the gold-rich crystalline structures of the Au/Cu⁰ nanocubes are close to 3.5 nm, while the copper nanoparticles are between 35–39 nm. The Rietveld refinement method was used to evaluate the crystalline phases and their quantities. The composition of Au/Cu⁰ is close to its nominal composition (Supplementary Table 2 and Supplementary Fig. 6). The XRD peak area at 40° with an index of Au(111) shows that the integrated peak areas in these Au/Cu⁰ nanocubes decrease with Au content. By controlling the concentration of Au, the diffraction intensity of the gold-rich crystalline structures decreases, and an $Au_{0.02}Cu_{0.98}$ in particular becomes very weak. In fact, high-resolution TEM clearly shows that single Au atoms are on the surface of the Cu⁰ nanocubes in $Au_{0.02}Cu_{0.98}$, which will be shown later.

Although atomic-resolution TEM can provide direct visualization of single-atom composition, it only provides information about a specific local area. In contrast, X-ray absorption spectroscopy (XAS) is element-sensitive and can be used to measure microcrystalline and amorphous materials with a large number of samples. XAS can be divided into X-ray absorption near-edge spectroscopy (XANES) and extended X-ray absorption fine structure (EXAFS). XANES is subjected to the measured elemental oxidation states and their local symmetries. In contrast, EXAFS can detect chemical bond distances (~0.01 Å level), coordination numbers, and structural disorders to detect the detailed local structure of elements. Unlike distinguishable single-atom composition, indistinguishable single-atom composition usually consists

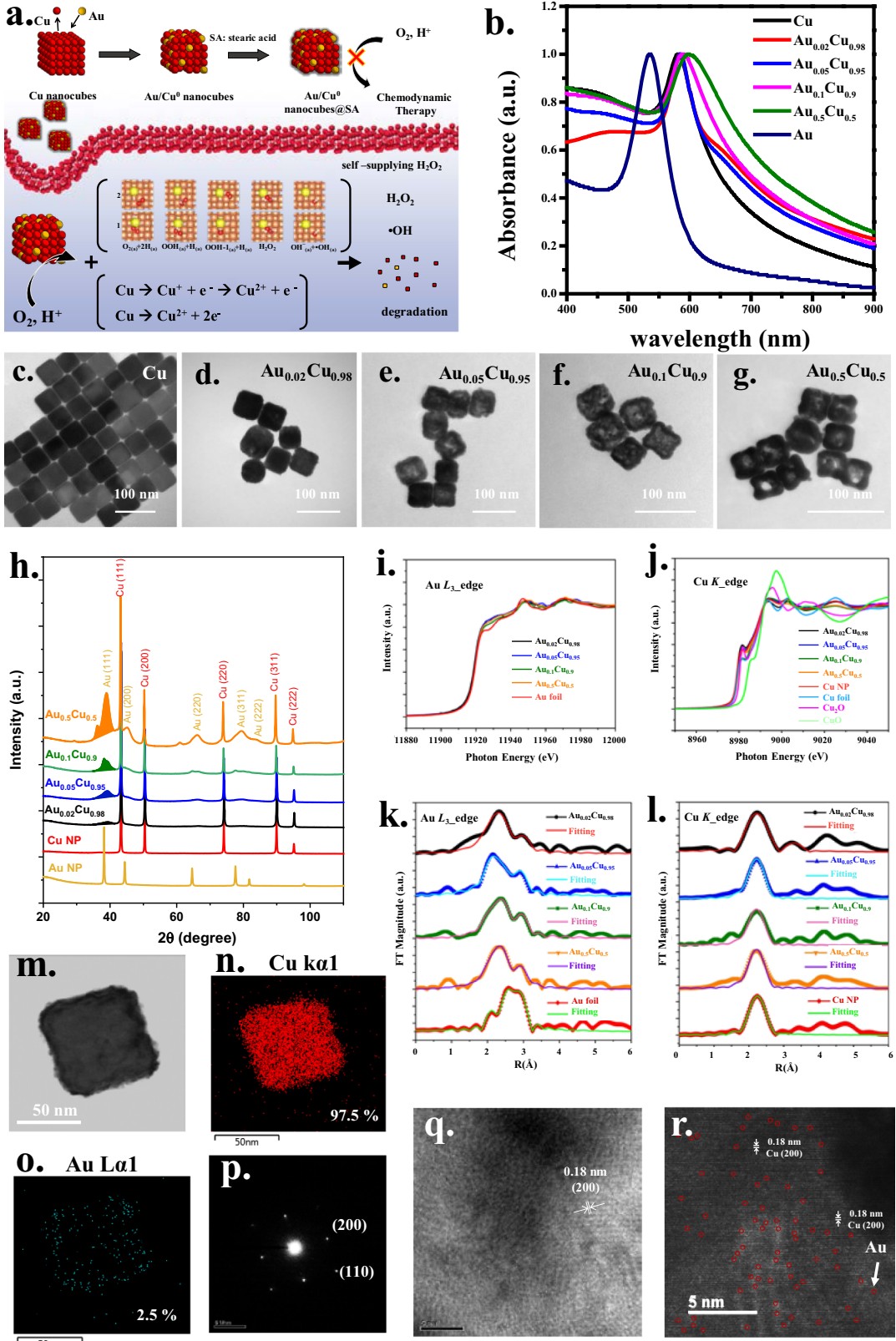

of two metallic elements with similar atomic numbers, or the two have close X-ray absorption energies. Thus, the characterization of such materials is very challenging[23–25]. We use a silicon drift detector (SDD) to collect energies at the Au $L\alpha$(9713 eV) in a 20 eV window, which allows for the elimination of fluorescence from Cu $K\alpha$, $K\beta$, and other scattered X-rays. The XANES spectra of Au $L_3$-edge and

Cu $K$-edge of Au/Cu⁰ nanocubes are respectively shown in Fig. 1i and j. Overlaid XANES spectra of Au/Cu⁰ nanocubes and metal foils indicate that Au and Cu are still the predominant oxidation states of zero valences. This suggests that the charges of Au and Cu are not changed after galvanic replacement. The Fourier-transformed EXAFS at the Au $L_3$-edge and Cu $K$-edge with model fits are,

**Fig. 1 | Characterization of Au/CuO nanocubes. a** The schematic diagram showing the fabrication of Au/Cu⁰ nanocubes using galvanic replacement and the generation of $H_2O_2$ and •OH. **b** UV–Vis profiles of Cu, Au, and different Au/Cu⁰ nanocubes. **c–g** TEM images of Cu nanocubes (solid morphology), $Au_{0.02}Cu_{0.98}$ (with a rough surface), $Au_{0.05}Cu_{0.95}$ (with slightly hollow appearance), $Au_{0.1}Cu_{0.9}$ (hollow structure), and $Au_{0.5}Cu_{0.5}$ (cage-like structure) under galvanic replacement reaction. (One representative data was shown from three independently repeated experiments). **h** Synchrotron X-ray powder diffraction patterns of Au/Cu⁰ nanocubes (colored online version). Peaks are indexed as Au (Fm-3m) and Cu (Fm-3m) crystallites. The entire Au(111) region colored under the peak near 40° indicates that the integrated peak areas in these Au/Cu⁰ nanocubes decrease with decreasing Au content. **i** and **j**. XANES spectra for Au $L_3$-edge and Cu $K$-edge for Au/Cu⁰ nanocubes including Cu NPs (nanocubes), Cu foil, $Cu_2O$, CuO, and Au foil. **k** and **l** Fourier-transformed EXAFS spectra for Au $L_3$-edge and Cu $K$-edge in Au/Cu⁰ nanocubes with the solid lines representing best-fit models. Spectra of Au foil and Cu NPs (nanocubes) for comparison. **m** and **o** HR-TEM image of the single $Au_{0.02}Cu_{0.98}$ nanocube and the corresponding atomic ratios in EDS mapping of Cu (red) and Au (cyan). **p** and **q** The diffraction pattern and lattice fringe of the $Au_{0.02}Cu_{0.98}$ nanocube. **r** AC HAADF-STEM image of the $Au_{0.02}Cu_{0.98}$ nanocube (red circles indicating Au single atoms). (**m–r** one representative data was shown from three independently repeated experiments).

respectively, shown in Fig. 1k and l. The detailed fitting results are summarized in Supplementary Table 3.

As shown in Fig. 1k and Table 1, the EXAFS spectrum of the Au $L_3$-edge shows that the Au/Cu⁰ nanocubes exhibit coordination shells at 2.64–2.7 Å, which is attributed to the Au–Cu bond in the first shell, while the second shell is the Au–Au bond at 2.82 Å. On the other hand, Fig. 1l and Table 1 show that the first shell of the Cu–Cu bond is located at 2.54 Å. The absence of Cu–Au coordination shells in the Cu $K$-edge EXAFS is due to the large Cu core nanocrystalline signal overshadowing the weak Cu–Au signal. This indicates that the galvanic replacement between Au atoms and Cu nanocubes is easily accomplished, leading to the formation of $Au_{0.02}Cu_{0.98}$ nanocubes. The coordination numbers (CN) of $Au_{0.02}Cu_{0.98}$ bonded to Cu and Au with Au as the central atom are, respectively, 4.73 and 6.87. These findings indicate the formation of Au atoms anchored on the surface of $Au_{0.02}Cu_{0.98}$ nanocubes. The bond distance of Au–Cu (2.67 Å) is longer than that of Cu–Cu (2.54 Å) and shorter than that of Au–Au (2.82 Å), which may create tensile-strained Cu atoms coordinated to Au atoms, thus boosting its chemical reactivity[26].

After an integral structural characterization in XRD, XANES, and EXAFS measurements, the $Au_{0.02}Cu_{0.98}$ specifically appeared to be an atomically dispersed Au onto Cu⁰ nanocube. Therefore, we further explore the features of $Au_{0.02}Cu_{0.98}$ using HR-TEM and aberration-corrected high-angle annular dark-field scanning transmission electron microscopy (AC-HAADF-STEM). As seen in Fig. 1m–o, the compositions of Au and Cu in $Au_{0.02}Cu_{0.98}$ could be, respectively, identified as 2.5% and 97.5% by EDX-mapping measurements, which is consistent with the AA determination. The $Au_{0.02}Cu_{0.98}$ reveals a well-fined crystalline with a lattice spacing of 0.18 nm corresponding to the (200) plane (Fig. 1p–r). Importantly, the AC-HAADF-STEM with the atomic resolution confirms atomically Au atoms (marked by red circles) bearing onto the Cu nanocube, as seen in Fig. 1r and the magnified view of Fig. 1r in Supplementary Fig. 7.

**Self-supplying $H_2O_2$ from aerobic $O_2$ and Fenton-like reaction**

Since the zero-valent Cu has been verified for the as-prepared nanocubes, the reduction potentials of copper (+0.522 V, $Cu^+/Cu$; +0.341 V, $Cu^{2+}/Cu$) are more negative than that (+0.695 V) of $O_2/H_2O_2$, making it possible for zero-valent Cu to thermodynamically reduce oxygen ($O_2$) to $H_2O_2$. A colorimetric method was used to measure $H_2O_2$ generation from the oxidation reaction between $H_2O_2$ and $KMnO_4$ by the observation of the drop of the absorption band at 525 nm contributed from $KMnO_4$ (Fig. 2a). The absorbance intensity of 525 nm decreased, indicating $H_2O_2$ generation, showing a different degree of color faded in colloidal solutions from the Au/Cu⁰ and pure Cu nanocubes, but no change in the Au nanocubes. The quantitative analysis of intensity from the response of $H_2O_2$ is summarized in Fig. 2b. The $Au_{0.02}Cu_{0.98}$ reveals superior $H_2O_2$ generation compared to other nanocubes. Next, we inspect whether the $H_2O_2$ generation is associated with aerobic or anaerobic conditions. A control examination is conducted to incubate $Au_{0.02}Cu_{0.98}$ nanocubes with $KMnO_4$ under $N_2$-filled (anaerobic) conditions. After 7 days of storage, we observed no change to color and absorption intensity at 525 nm, consistent only with the $KMnO_4$ (Fig. 2c and Supplementary Fig. 8). Therefore, the $H_2O_2$ is in situ generated from the $O_2$ in the aerobic environment. The quantitative results of $H_2O_2$ were shown in Fig. 2d. The $H_2O_2$ generation was evaluated by hydrogen peroxide assay kit in different compositions of $Au_xCu_y$ (Cu, $Au_{0.02}Cu_{0.98}$, $Au_{0.05}Cu_{0.95}$, $Au_{0.1}Cu_{0.9}$, $Au_{0.5}Cu_{0.5}$, and Au). The largest $H_2O_2$ generation was calculated to be 122 μM in $Au_{0.02}Cu_{0.98}$ in 10 min of reaction. The amount of $H_2O_2$ in $Au_{0.02}Cu_{0.98}$ is higher than the endogenous $H_2O_2$ in tumoral microenvironments (~100 μM) and can be generated in the presence of $O_2$. The $Au_{0.02}Cu_{0.98}$, which has the greatest capability to form $H_2O_2$, was then chosen to demonstrate time- and concentration-dependence (Fig. 2e and Supplementary Fig.9). Because of the generation of $H_2O_2$, Cu is a potential candidate to process Fenton-like reaction for •OH formation. The •OH generation was identified by the increased fluorescence emission ($\lambda_{em} = 425$ nm)

## Table 1 | EXAFS fitting results of Au $L_3$_edge and Cu $K$_edge for Au/Cu⁰ nanocubes[a]

| sample | Center atom | Neighboring atom CN | | | | Bond distance (R, Å) | |
|---|---|---|---|---|---|---|---|
| | | Cu | Au | CN_total | χ (%) | Cu | Au |
| $Au_{0.02}Cu_{0.98}$ | Au | 4.73(2) | 6.87(1) | 11.60(2) | 42 | 2.67(9) | 2.82(3) |
| $Au_{0.05}Cu_{0.95}$ | Au | 2.43(1) | 6.55(1) | 8.98(2) | 27 | 2.64(6) | 2.82(3) |
| $Au_{0.1}Cu_{0.9}$ | Au | 2.27(2) | 7.45(1) | 9.72(2) | 23 | 2.69(11) | 2.83(3) |
| $Au_{0.5}Cu_{0.5}$ | Au | 1.46(1) | 7.53(1) | 8.99(1) | 16 | 2.70(1) | 2.81(4) |
| Au foil | Au | 0 | 12 | 12 | 0 | 0 | 2.85(2) |
| $Au_{0.02}Cu_{0.98}$ | Cu | 5.85(1) | 0 | 5.85(1) | 0 | 2.54(1) | 0 |
| $Au_{0.05}Cu_{0.95}$ | Cu | 4.92(1) | 0 | 4.92(1) | 0 | 2.54(4) | 0 |
| $Au_{0.1}Cu_{0.9}$ | Cu | 4.14(1) | 0 | 4.14(1) | 0 | 2.53(1) | 0 |
| $Au_{0.5}Cu_{0.5}$ | Cu | 3.96(1) | 0 | 3.96(1) | 0 | 2.53(1) | 0 |
| Cu NP | Cu | 6.1(4) | 0 | 6.1(4) | 0 | 2.54(1) | 0 |
| Cu foil | Cu | 12 | 0 | 12 | 0 | 2.54(1) | 0 |

[a]χ: the extent of heteroatomic intermix for neighboring atom; CN: coordination number; CN_total: CN around center atom; R: bond distance.

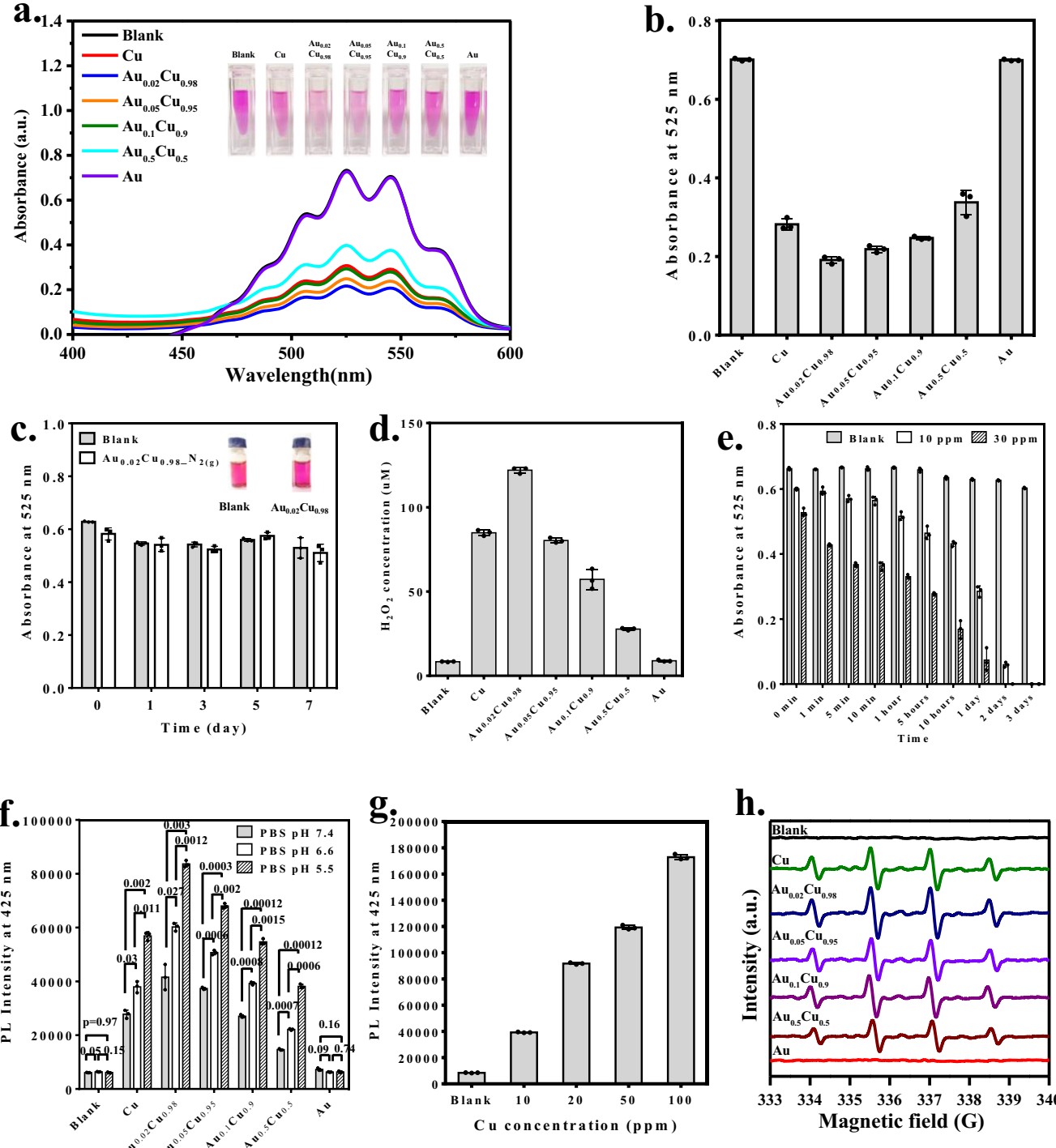

**Fig. 2 | H₂O₂, and ·OH generation from Au/CuO nanocubes. a** UV–Vis profile derived from the colorimetric analysis showing the change of absorbance with Cu, Au, and different Au/Cu⁰ nanocubes (the inset showing the response of H₂O₂ given the change of colloidal color). (One representative data was shown from three independently repeated experiments). **b** The quantitative analysis by the response of H₂O₂ for Cu, Au, and Au/Cu⁰ nanocubes determined the absorbance intensity at 525 nm. **c** The quantitative analysis by the response of H₂O₂ in KMnO₄ only (blank) or KMnO₄ with Au₀.₀₂Cu₀.₉₈ at 525 nm absorbance under N₂ as a function of time (inset showing colloidal color after 7-day incubation). **d** Quantitative detection of H₂O₂ production yield from hydrogen peroxide assay kit showing fluorescence emission (λₑₘ = 510 nm) after reaction with Cu, Au, and Au/Cu⁰ nanocubes. **e** The quantitative analysis by the response of H₂O₂ in Au₀.₀₂Cu₀.₉₈ as a function of time and concentration. **f** The ·OH generation detected by the fluorescence emission (λₑₘ = 425 nm) using terephthalic acid (TPA) probe under different pH values. **g** The quantitative analysis of ·OH generation in Au₀.₀₂Cu₀.₉₈ by the fluorescence intensity at 425 nm under different concentrations at pH 5.5. **h** The 1:2:2:1 amplitude with quartet ESR signals of DMPO−OH associated with ·OH from Cu, Au, and different Au/Cu⁰ nanocubes. All data were obtained in triplicate (n = 3, the error bars represented mean ± SD, p-values were calculated by one-way ANOVA. Source data are provided as a Source Data file).

of the terephthalic acid (TPA) probe in different pH values (Fig. 2f). More •OH can be produced by $Au_{0.02}Cu_{0.98}$ than other groups of Au/$Cu^0$ nanocubes. Specifically, a significant enhancement can be observed in acid conditions. The •OH increased as a function of $Au_{0.02}Cu_{0.98}$ concentration at pH 5.5 (Fig. 2g). The amplitude of the electron spin resonance (ESR) signal evidenced more •OH generation in $Au_{0.02}Cu_{0.98}$ among the Au/$Cu^0$ nanocubes (Fig. 2h). According to the above results, the $Au_{0.02}Cu_{0.98}$ could act as a promising chemodynamic agent following $O_2 \rightarrow H_2O_2 \rightarrow$ •OH reactions.

## The degradable nature of the $Au_{0.02}Cu_{0.98}$ nanocubes

Since we have observed the reactions of $O_2 \rightarrow H_2O_2 \rightarrow$ •OH, the zero-valent Cu ($Cu^0$) is likely oxidized to $Cu^+$ and $Cu^{2+}$. The nanocubes may release oxidized copper ions resulting in the dissolution of structures. The stability of $Au_{0.02}Cu_{0.98}$ nanocubes showing superior $H_2O_2$ and •OH generation was then monitored for 7 days under $H_2O$, PBS (pH 7), and PBS (pH 5.5) (Fig. 3a). Under PBS, nanocubes already started to decompose at day 0. Higher acidity resulted in faster dissolution, with nanocubes nearly disintegrating completely after 1-day incubation. Under an $H_2O$ environment, the structures also showed apparent dissolution after 3-day incubation. This self-decomposing behavior makes the $Au_{0.02}Cu_{0.98}$ nanocubes a promising renal-clearable agent when applied in in vivo for tumor studies. We further used XPS measurements to inspect the signal of $Cu^{2+}$ formation by incubation of $Au_{0.02}Cu_{0.98}$ nanocubes in PBS (pH 7) after 1 day. The intensity of $Cu^{2+}$ increased post-1 day given the evidence of $Cu^0$ oxidized to $Cu^{2+}$(Fig. 3b). On the one hand, the degradable nature under acidic conditions favors excretion through urinary metabolism. On the other hand, the self-decomposing behavior may also cause oxidative stress in the course of the blood circulation in vivo because of •OH generation. To avoid the self-generation of $H_2O_2$ during blood circulation, the stearic acid (SA) molecule was then used to modify the surface of the nanocubes. The modification of SA retained the structure of $Au_{0.02}Cu_{0.98}$ nanocubes resulting in the dispersed colloidal solutions (Fig. 3c). A layer of SA coated on the nanocube can be clearly seen (inset of Fig. 3c). FTIR analysis provided evidence of SA on the surface of $Au_{0.02}Cu_{0.98}$ nanocubes (Supplementary Fig. 10). To further support the successful SA coating on nanocubes, the generation of $H_2O_2$ (Fig. 3d) and •OH (Fig. 3e) was, respectively, conducted using $KMnO_4$ treatment and TPA fluorescence probe. Neither a drop in intensity from the $KMnO_4$ treatment nor fluorescence from the TPA probe was detected from $Au_{0.02}Cu_{0.98}$@SA. The generation of $H_2O_2$ and •OH was successfully suppressed after SA modification. In addition, the $Au_{0.02}Cu_{0.98}$@SA nanocubes exhibited excellent stability and retained their morphology features under different solution conditions (Fig. 3a). No dissolution of the nanocubes was seen after 7-day incubation.

## Simulation analysis for oxidase and Fenton-like reactions

To gain more insight into the mechanisms by which Au/$Cu^0$ nanocubes proceed with the reactions of $O_2 \rightarrow H_2O_2 \rightarrow$ •OH, density functional theory (DFT) was carried out to calculate the reaction barriers of hydrogen peroxide formation via $O_2$ hydrogenation on Au/$Cu^0$ nanocubes with different Au/Cu ratios, including pure Cu, $Au_{0.02}Cu_{0.98}$, $Au_{0.5}Cu_{0.5}$, and pure Au. The (100) surface model was considered for these Au/$Cu^0$ nanocubes, and the surfaces were pre-covered with hydrogen atoms. The water dehydrogenation reaction to produce the hydrogen atoms was also calculated. As shown in Supplementary Table 4 and Supplementary Fig. 11, the first dehydrogenation barriers of $H_2O$ are similar on pure Cu, $Au_{0.02}Cu_{0.98}$, and $Au_{0.5}Cu_{0.5}$, but they are much larger on pure Au. The second dehydrogenation barriers of the OH group on pure Cu and $Au_{0.02}Cu_{0.98}$ are smaller than those on $Au_{0.5}Cu_{0.5}$ and pure Au. It reveals that the hydrogen formation rates on Cu and $Au_{0.02}Cu_{0.98}$ are faster than those on $Au_{0.5}Cu_{0.5}$ and pure Au. Besides, we have also calculated the adsorption energies of the $H_2O$

molecule, OH group, H atom, and O atom on all possible active sites at the Cu(100), $Au_{0.02}Cu_{0.98}$(100), $Au_{0.5}Cu_{0.5}$(100), and Au(100) surfaces, as shown in Supplementary Figs. 11–15 and Supplementary Tables 4–8. The intermediates in the reaction mechanisms were carried out using the most stable adsorption structures to proceed. All the adsorption energies, activation energies, and reaction energies were considered using the solvent effect.

Supplementary Figs. 16, 17 and Supplementary Table 9 show the side-on and end-on configurations of $O_2$ molecules on all the possible active sites at Cu(100), $Au_{0.02}Cu_{0.98}$(100), $Au_{0.5}Cu_{0.5}$(100), and Au(100) surfaces. According to the calculated adsorption energies, all the results show that the side-on configuration is more stable than the end-on configuration for the $O_2$ molecule. The calculated most stable adsorption energies of $O_2$ on pure Cu, $Au_{0.02}Cu_{0.98}$, $Au_{0.5}Cu_{0.5}$, and pure Au are, respectively, −2.12, −2.00, −1.16, and −0.42 eV. These results indicate that the interaction of $O_2$ to the surface is proportional to the number of copper atoms, meaning that the gold atoms would reduce the activity of the copper atoms in the alloys. In addition, as shown in Fig. 4a, the calculated hydrogenation barriers from $O_2$ to OOH and OOH to $H_2O_2$ on pure Cu are 1.32 and 1.29 eV, respectively. For the $Au_{0.02}Cu_{0.98}$, there are two different reaction pathways via two chemical environments: (1) with the Au atom and (2) without the Au atom (Fig. 4b). The hydrogenation barriers of two-step reactions in pathway 1 are 0.75 and 0.76 eV, while those in pathway 2 are 0.98 and 0.99 eV. For both pathways, the activation energy of the hydrogenation reaction on the $Au_{0.02}Cu_{0.98}$ is smaller than that of Cu. For the $Au_{0.5}Cu_{0.5}$, the two hydrogenation barriers for forming the $H_2O_2$ molecule from $O_2$ are 1.15 and 0.61 eV (Fig. 4c). Figure 4d shows the hydrogenation barriers from $O_2$ to OOH and OOH to $H_2O_2$ are, respectively, 0.48 and 0.74 eV on pure Au. Although the activation energies are small on both $Au_{0.5}Cu_{0.5}$ and pure Au, the first hydrogenation barrier is larger than the adsorption energy of the $O_2$ molecule. That is, the reaction of the first hydrogenation on these two surfaces is more difficult, resulting in the relatively poor reactivity of $O_2$ hydrogenation on both $Au_{0.5}Cu_{0.5}$ and pure Au.

Furthermore, the reaction energies between the $H_2O_2$ desorption and the production of OH radicals were also performed. For the production of •OH radical, we carried out a Fenton-like reaction as follows: $H_2O_{2(a)} \rightarrow OH^-_{(a)} + \bullet OH_{(aq)}$ ; one •OH radical can form in the solvent via desorption and the other OH group would adsorb on the surface to form $OH^-$ anion. The $OH^- + \bullet OH$ can be clarified by the projected density of states (PDOS). As shown in Supplementary Fig. 18, the peaks of the adsorbed OH group have no spin-splitting property, while the gas-phase OH group possesses the spin-splitting character. This reveals that the electron configuration of the adsorbed OH group is fully-filled, but the electron configuration of the gas-phase OH group is half-filled. Thus, the adsorbed OH and the gas-phase OH group can be regarded as OH anion and •OH radical, respectively. On the pure Cu, the desorption energy of $H_2O_2$ is 0.35 eV while the reaction energy of •OH radical is −0.03 eV. For the $Au_{0.02}Cu_{0.98}$, the desorption energy of $H_2O_2$ is 0.34 eV, whereas the reaction energy of •OH radical formation is −0.01 eV. The •OH radical formation energies are less than those of $H_2O_2$ desorption on both pure Cu and $Au_{0.02}Cu_{0.98}$, indicating that the •OH radical formation readily occurs on these two nanocubes. Contrarily, the desorption energy of $H_2O_2$ is smaller than that of •OH radical formation on either $Au_{0.5}Cu_{0.5}$ or pure Au, reflecting that the •OH radical production on these two Au/$Cu^0$ nanocubes is thermodynamically unfavorable. Overall, DFT results reveal that the kinetic effect of $H_2O_2$ formation on $Au_{0.02}Cu_{0.98}$ is faster than on other compositions of nanocubes, and the thermodynamic effect of •OH radical production is favorable on $Au_{0.02}Cu_{0.98}$. All the calculated reaction barriers and reaction energies of $O_2$ hydrogenation to $H_2O_2$ and •OH formation are listed in Table 2. The reactivity of •OH radical production on $Au_{0.02}Cu_{0.98}$ is superior to pure Cu, $Au_{0.5}Cu_{0.5}$, and pure Au.

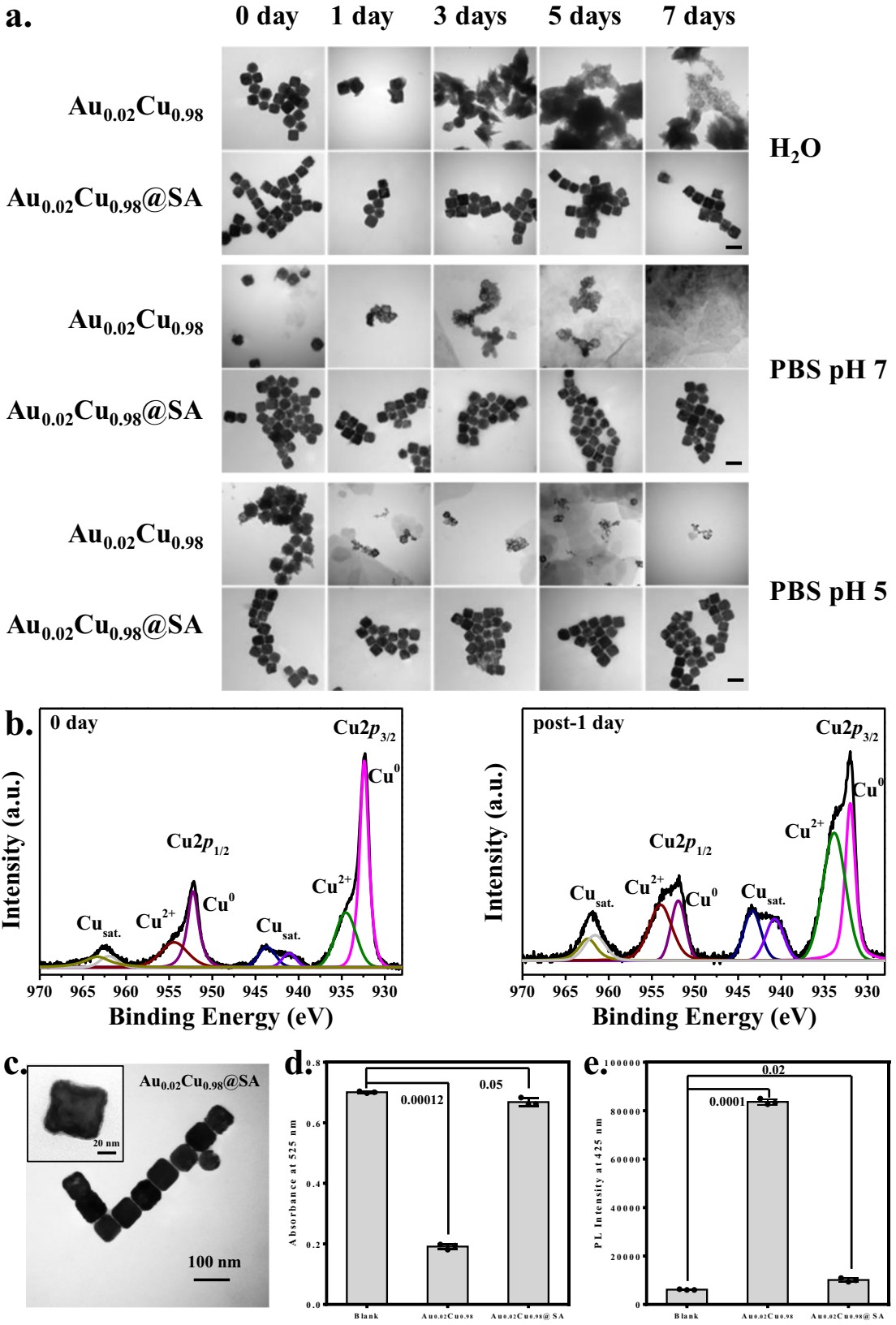

Figure 4e–h shows the electron localization function (ELF) diagrams of the pure Cu, $Au_{0.02}Cu_{0.98}$, $Au_{0.5}Cu_{0.5}$, and pure Au. The electrons locate around the hollow position between the copper atoms on the pure Cu (Fig. 4e), reflecting the strong activity of the hollow site for the pure Cu. On the other hand, the electron localization around the hollow site becomes weaker while the electrons begin to accumulate more to the gold atoms from $Au_{0.02}Cu_{0.98}$ to $Au_{0.5}Cu_{0.5}$ (Fig. 4b and c). For the pure Au (Fig. 4d), the ELF plot displays the electron localization around the gold atoms rather than at the hollow region. Since the electronegativity of Au is larger than the Cu, the charge of the Au atoms can become more negative in the $Au/Cu^0$ nanocubes. According to previous theoretical studies[27,28], increased

**Fig. 3 | The stability and self-decomposition behavior in $Au_{0.02}Cu_{0.98}$ and $Au_{0.02}Cu_{0.98}$@SA under different conditions ($H_2O$, PBS at pH = 5.5 and 7). a** TEM images revealed that the $Au_{0.02}Cu_{0.98}$ nanocubes started to decompose within 1 day in PBS and $H_2O$. Contrarily, the $Au_{0.02}Cu_{0.98}$@SA remained intact structures over 7 days of incubation under all conditions (all scale bars as 100 nm). **b** The XPS spectrum of $Au_{0.02}Cu_{0.98}$ nanocubes given $Cu^0$ and $Cu^{2+}$ signals under PBS condition (pH 7) as a function of the day ($2p_{3/2}$ and $2p_{1/2}$ assigned as 932 and 952 eV, respectively, for $Cu^0$ and assigned as 934 and 954 eV, respectively, for $Cu^{2+}$). **c** TEM image of the $Au_{0.02}Cu_{0.98}$@SA nanocubes (inset: a transparent layer surrounded by $Au_{0.02}Cu_{0.98}$ surface indicating the presence of SA). **d** and **e** Quantitative analysis of $H_2O_2$ and ·OH generation in $Au_{0.02}Cu_{0.98}$@SA nanocubes through $KMnO_4$ and TPA fluoresce probe treatments, respectively. The suppression of $H_2O_2$ and •OH due to the SA modification. All data were obtained in triplicate ($n = 3$, The error bars represented mean ± SD, $p$-values were calculated by one-way ANOVA. Source data are provided as a Source Data file.) (**a**, **c** one representative data was shown from three independently repeated experiments).

negative charge distribution of Au atoms will reduce the activity of bimetallic systems. As there is small amount of Au atoms in $Au_{0.02}Cu_{0.98}$, the activity of $Au_{0.02}Cu_{0.98}$ is anticipated to slightly decrease, thus facilitating the reactivity of the $O_2$ hydrogenation to $H_2O_2$. As a reference, our aforementioned XANES measurements indicate the charges of Au and Cu do not change after galvanic replacement. Generally, the association reaction between the reactant (i.e. $O_2$ here) and hydrogen atoms would be enhanced as the binding activity gradually decreases between the catalyst (i.e. Cu here) and both reactant and hydrogen atoms[28]. That is, the small amount of Au atoms in $Au_{0.02}Cu_{0.98}$ reduces Cu activity in oxygen binding, which in turn strengthens the hydrogenation reaction. In comparison with pure Cu, the adsorption activities of $O_2$ and H atoms are weakened by small amounts of Au atoms in $Au_{0.02}Cu_{0.98}$ and then the catalytic property of $O_2$ reduction to $H_2O_2$ is enhanced at the $Au_{0.02}Cu_{0.98}$. As the concentrations of Au atoms in the $Au/Cu^0$ nanocubes gradually increase to the $Au_{0.5}Cu_{0.5}$ and pure Au, the binding strengths to $O_2$ are significantly decreased. However, this change weakens the overall reactivity of both $Au_{0.5}Cu_{0.5}$ and pure Au because the hydrogenation barriers exceed the adsorption energy of the $O_2$ on both $Au_{0.5}Cu_{0.5}$ and pure Au. Therefore, although the binding strength to $O_2$ can be decreased on both $Au_{0.5}Cu_{0.5}$ and pure Au, the large amount of Au atoms limits, rather than facilitates, the reaction.

The mechanisms by which of the degradable $Au/Cu^0$ nanocubes to trigger the self-supply of $H_2O_2$ followed by the Fenton-like reaction are then proposed to perform CDT. As shown in Fig. 1a, the different ratios of $Au/Cu^0$ nanocubes could be synthesized through galvanic replacement. Once $Au_{0.02}Cu_{0.98}$ nanocubes proceed the endocytosis, the nanocubes can act as catalysts for oxygen hydrogenation and the subsequent Fenton-like reaction. Concurrently, $Au_{0.02}Cu_{0.98}$ begins to oxidize from $Cu^0$ to $Cu^+$ and/or $Cu^{2+}$ ($Cu \rightarrow Cu^+ + e^- \rightarrow Cu^{2+} + 2e^-$). Evidence for the presence of $Cu^+$ in cells will be provided later using a CopperGreen dye, a $Cu^+$ indicator. The released electrons are then used to reduce $O_2$ to $H_2O_2$ under acidic conditions ($O_2 + 2H^+ + 2e^- \rightarrow H_2O_2$). Therefore, the catalytic reactions of $Au_{0.02}Cu_{0.98}$ and the formation of the released $Cu^+$ or $Cu^{2+}$ simultaneously contribute to the generation of $H_2O_2$ and •OH. The released copper ions cause the nanocubes to decompose, which favor metabolic excretion.

## In vitro evaluation
For in vitro studies, all Cu, $Au/Cu^0$, and Au nanocubes were modified with SA. MTT assay was used to investigate the cell viability of HepG2-Red-FLuc hepatocellular carcinoma cells incubated with Cu@SA, Au/$Cu^0$@SA, and Au@SA nanocubes where $Au_{0.02}Cu_{0.98}$@SA outperformed other groups in killing cancer cells (Fig. 5a). Fluorescence staining experiments in live and dead cells also demonstrated the efficacy of $Au_{0.02}Cu_{0.98}$@SA in damaging cancer cells (Fig. 5b). Decreased cell survival rate was observed as a function of $Au_{0.02}Cu_{0.98}$@SA concentrations (Fig. 5c). Flow cytometry analysis demonstrated the increased late-apoptosis after 24 h incubation in $Au_{0.02}Cu_{0.98}$@SA nanocubes (Fig. 5d). The cells incubated with $Au_{0.02}Cu_{0.98}$@SA have a relatively higher late apoptotic ratio (39.09%) compared to other groups (cell only: 0%, Cu@SA: 16.43%, $Au_{0.05}Cu_{0.95}$@SA: 14.24%, $Au_{0.1}Cu_{0.9}$@SA: 5.5%, $Au_{0.5}Cu_{0.5}$@SA: 0.17%, and Au@SA: 0.42%). The decomposition of nanocubes was again monitored by the morphological change when nanocubes were cultured with cancer cells as a function of time (Fig. 6a). TEM images displayed the structural disintegration of $Au_{0.02}Cu_{0.98}$@SA nanocubes with cells as a function of time as compared with nanocubes with no cell incubation. CopperGreen dye with green emission was used to verify the presence of oxidized $Cu^+$ ions in cells (Fig. 6b). In addition, hydrogen peroxide assay kit and APF fluorescence agent were, respectively, used to confirm the signals of $H_2O_2$ and •OH produced from $Au_{0.02}Cu_{0.98}$@SA in cells (Fig. 6c, d). Contrarily, the very weak fluorescence signals were seen in anoxic condition (2% $O_2$) corresponding to the low amount of $H_2O_2$ and •OH generation, indicating the support of the $O_2 \rightarrow H_2O_2 \rightarrow$ •OH reactions under $Au_{0.02}Cu_{0.98}$@SA treatment (Supplementary Fig. 19). No hemolysis and damage in vascular endothelial cells were observed from $Au_{0.02}Cu_{0.98}$@SA (Fig. 5e, f), which ensures safety in the course of blood circulation.

## In vivo animal studies against tumors
To validate the in vivo biosafety of nanocubes, the C57BL/6 mice were intravenously administered with 600 ppm $Au_{0.02}Cu_{0.98}$@SA (in sterilized PBS) and continuously observed for 7 days. There was no significant difference in the body weight, blood biochemical index (for liver and kidney functions), and histological morphology (heart, lung, liver, spleen, kidney) between the nanocubes group and PBS control group, which suggests our $Au_{0.02}Cu_{0.98}$@SA nanocubes do not have acute toxicity. (Supplementary Figs. 20, 21 and Fig. 7a). In vivo biodistribution indicates that, among major organs, nanocubes accumulation concentrates in the liver, but gradually decreased as a function of time. However, the Cu content in urine increased post-24 h (Fig. 7b), which is associated with the decomposition of $Au_{0.02}Cu_{0.98}$@SA to release copper ions. The amount of copper accumulation in urine was 10 times greater than that of the PBS post-24 h, indicating the clearance effect. The inset of Fig. 7b has removed liver results to clarify the enhanced accumulation in urine.

The in vivo antitumor efficacy was established in the HepG2-Red-FLuc orthotopic hepatocellular carcinoma xenograft mice. The tumor-bearing mice were randomly treated with one dose of 600 ppm nanocubes including Au@SA, $Au_{0.02}Cu_{0.98}$@SA, $Au_{0.5}Cu_{0.5}$@SA, and Cu@SA as well as PBS (as control) through intravenous administration, respectively. The Cu@SA but not Au@SA shows the anti-tumor effect observed from the IVIS system (Fig. 7c), which is consistent with the in vitro study. Surprisingly, the one-dose treatment with $Au_{0.02}Cu_{0.98}$@SA exhibited more remarkable antitumor efficacy, showing significant inhibition of tumor growth, than the Cu@SA. On the other hand, the different atomic ratio of $Au_{0.5}Cu_{0.5}$@SA cannot suppress tumor growth, indicating the significant efficacy of $Au_{0.02}Cu_{0.98}$@SA (Fig. 6c). The tumoral sizes (Supplementary Fig. 22a) and the total flux by IVIS (Fig. 6d) of the remaining hepatocellular carcinoma treated with different nanocubes were analyzed after mice sacrificed. Among the treatments, $Au_{0.02}Cu_{0.98}$@SA was confirmed to possess the best anti-tumor effect with tumor nearly suppressed. We also examined the morphology change in tumors, and found that $Au_{0.02}Cu_{0.98}$@SA treatment induced more tumor necrosis (area outside the white dots enclosed region in each picture) than other treated groups (Supplementary Fig. 22b). Moreover, the abundant nuclear expression of γ-H2AX (DNA damage marker) and cytoplasmic expression of cleaved caspase-3 (apoptosis cell death marker) were observed within the tumor region because of the excessive ROS

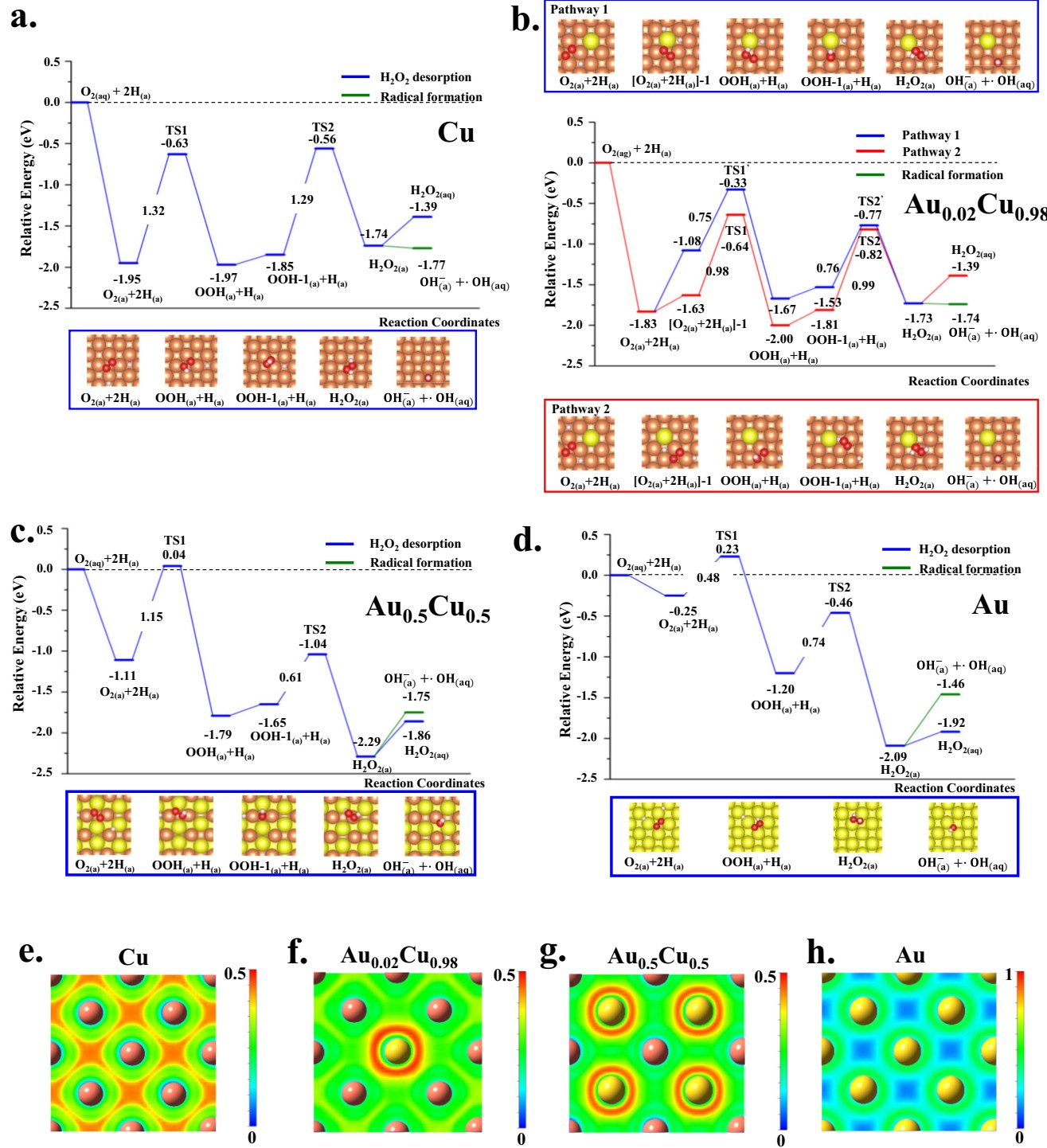

**Fig. 4 | Simulation analysis in Cu, Au, and different Au/CuO.** The calculated potential energy profiles for the $O_2$ reduction to $H_2O_2$ on the **a** pure Cu, **b** $Au_{0.02}Cu_{0.98}$, **c** $Au_{0.5}Cu_{0.5}$, and **d** pure Au. The calculated electron localization function (ELF) diagrams of the **e** pure Cu, **f** $Au_{0.02}Cu_{0.98}$, **g** $Au_{0.5}Cu_{0.5}$, and **h** pure Au. Brown, gold, red and white spheres represent Cu, Au, O, and H atoms, respectively.

production by $Au_{0.02}Cu_{0.98}$@SA treatment in tumors (Fig. 6e). Taken together, these results indicate the excellent efficacy of $Au_{0.02}Cu_{0.98}$@SA for antineoplastic therapy in hepatocellular carcinoma. Evaluation of the clinical potential is in progress.

We used galvanic replacement to fabricate different compositions of zero-valent Cu nanocubes which generated $H_2O_2$ and •OH by taking $O_2$ from air under aerobic conditions. Specifically, $Au_{0.02}Cu_{0.98}$ outperformed other $Au/Cu^0$ nanocubes in the production of $H_2O_2$ and •OH. The degradable $Au_{0.02}Cu_{0.98}$ released $Cu^+$ and $Cu^{2+}$ resulting in

oxygen reduction and Fenton-like reactions. Simulation analysis also revealed the catalytic capability of $Au_{0.02}Cu_{0.98}$ with the facilitation of Au for $O_2 \rightarrow H_2O_2 \rightarrow$ •OH reactions.

## Methods

### Chemicals

All reagents were in analytical purity and used without further purification. Ethanol ($C_2H_5OH$, 99.9%) was purchased from J.T. Baker. Aminophenyl fluorescein solution (APF, $C_{26}H_{17}NO_5$, 98%) was

**Table 2 | Calculated reaction barriers ($E_a$ in eV) and reaction energies ($\Delta E$ in eV) for elementary reactions of $O_2$ hydrogenation to $H_2O_2$ on the Au/Cu$^0$ nanocubes**

| Elementary steps | $E_a$ | $\Delta E$ |
|---|---|---|
| **Cu(100)** | | |
| $O_{2(a)} + H_{(a)} \rightarrow OOH_{(a)}$ | 1.32 (1.31) | −0.02 (0.04) |
| $OOH_{(a)} \rightarrow H_2O_{2(a)}$ | 1.29 (1.31) | 0.11 (0.22) |
| $H_2O_{2(a)} \rightarrow OH^-_{(a)} + \bullet OH_{(aq)}$ | | −0.03 (−0.04) |
| $H_2O_{2(a)} \rightarrow H_2O_{2(aq)}$ | | 0.35 (0.46) |
| **Au$_{0.02}$Cu$_{0.98}$** | | |
| (Pathway 1) $O_{2(a)} + H_{(a)} \rightarrow OOH_{(a)}$ | 0.98 (0.96) | −0.37 (−0.31) |
| (Pathway 1) $OOH_{(a)} \rightarrow H_2O_{2(a)}$ | 0.99 (1.08) | 0.08 (0.21) |
| (Pathway 2) $O_{2(a)} + H_{(a)} \rightarrow OOH_{(a)}$ | 0.75 (0.76) | −0.59 (−0.57) |
| (Pathway 2) $OOH_{(a)} \rightarrow H_2O_{2(a)}$ | 0.76 (0.72) | −0.20 (−0.17) |
| $H_2O_{2(a)} \rightarrow OH^-_{(a)} + \bullet OH_{(aq)}$ | − | −0.01 (0.03) |
| $H_2O_{2(a)} \rightarrow H_2O_{2(aq)}$ | | 0.34 (0.42) |
| **Au$_{0.5}$Cu$_{0.5}$** | | |
| $O_{2(a)} + H_{(a)} \rightarrow OOH_{(a)}$ | 1.15 (1.13) | −0.68 (−0.56) |
| $OOH_{(a)} \rightarrow H_2O_{2(a)}$ | 0.61 (0.63) | −0.64 (−0.64) |
| $H_2O_{2(a)} \rightarrow OH^-_{(a)} + \bullet OH_{(aq)}$ | | 0.54 (0.64) |
| $H_2O_{2(a)} \rightarrow H_2O_{2(aq)}$ | | 0.42 (0.62) |
| **Au(100)** | | |
| $O_{2(a)} + H_{(a)} \rightarrow OOH_{(a)}$ | 0.48 (0.48) | −0.95 (−0.87) |
| $OOH_{(a)} \rightarrow H_2O_{2(a)}$ | 0.74 (0.81) | −0.89 (−0.80) |
| $H_2O_{2(a)} \rightarrow OH^-_{(a)} + \bullet OH_{(aq)}$ | − | 0.63 (0.68) |
| $H_2O_{2(a)} \rightarrow H_2O_{2(aq)}$ | | 0.17 (0.31) |

The values outside and inside the parentheses represent with and without considering the solvent effect.

acquired from Life Technologies. A hydrogen peroxide assay kit was acquired from abcam. CopperGreen dyes were acquired from Merck. Copper(I) bromide (CuBr, 98%), Octadecylamine ($CH_3(CH_2)_{17}NH_2$, 99%), trioctylphosphine oxide ($[CH_3(CH_2)_7]_3PO$, 90%), Oleylamine ($CH_3(CH_2)_7CH=CH(CH_2)_7CH_2NH_2$, 90%), cetyltrimethylammonium bromide ($C_{19}H_{42}BrN$), Polyvinylpyrrolidone (PVP, $(C_6H_9NO)_n$, MW = 55,000), hydrogen peroxide solution ($H_2O_2$, 30%), Sodium bromide (NaBr, 99.5%), ascorbic acid ($C_6H_8O_6$, 99%), cetyltrimethylammonium chloride ($C_{19}H_{42}ClN$, 25%), Sodium borohydride (NaBH$_4$, 99%) and 3-(4,5-dimethylthiazol-2-yl)−2,5-diphenyltetrazolium bromide (MTT, $C_{18}H_{16}BrN_5S$, 97.5%) were bought from Sigma-Aldrich. Water was obtained by using a Millipore direct-Q deionized water system throughout all studies.

### Preparation of Cu nanocubes
A total of 0.05 g of CuBr, 0.08 g of octadecylamine (ODA), and 1 g of trioctylphosphine oxide (TOPO) were mixed in 20 mL of oleylamine. Subsequently, the reaction apparatus was filled with argon, heated at 20 °C/min in a heating jacket, and maintained at 300 °C for 10 min until the Cu nanoparticles (NPs) grown into nanocubes. The reaction solution was cooled to room temperature and centrifuged at 6200 × g for 5 min. After the supernatant was removed, the nanocubes were washed three times using a toluene solution. Finally, Cu nanocubes were collected and stored in oleylamine.

### Preparation of Au/Cu$^0$ nanocubes
The cetyltrimethylammonium bromide (CTAB) and polyvinylpyrrolidone (PVP) solutions emulsified the Cu nanocubes from the oil phase to the water phase. And then, Cu nanocubes acted as a sacrificial template of the galvanic replacement reaction, and HAuCl$_4$ solution was used as the precursor of the reaction. First, the Cu nanocubes were distributed in a 100-μL toluene solution with a

concentration of 10,000 ppm. Subsequently, 10 mL of CTAB and PVP solution was added and shaken evenly, and the Cu nanocubes were emulsified from the oil phase to the water phase and dispersed in the aqueous solution. Various volumes of HAuCl$_4$ solution with a molar concentration of 50 mM were added to systematically alter the composition of the Au/Cu$^0$ nanocubes. The Au/Cu$^0$ nanocubes with an Au:Cu ratio of 0.02:0.98, 0.05:0.95, 0.1:0.9, and 0.5:0.5 were obtained after 5, 10, 20, and 40 μL of HAuCl$_4$ were added, respectively. The alloyed Au$_x$Cu$_y$ nanocages (Au$_{0.75}$Cu$_{0.25}$), nanoframes (Au$_{0.8}$Cu$_{0.2}$), and completed replacement Au nanoparticles were obtained after 60, 80, and 100 μL of 50 mM HAuCl$_4$ treatments, respectively.

### Preparation of Au nanocubes
Gold seeds were prepared first by volume of 10 mL $H_2O$ containing 1325 μL cetyltrimethylammonium chloride (CTAC) (25%), 500 μL of 5 mM HAuCl$_4$ solution, and 450 μL of 0.02 M NaBH$_4$. Next, two vials were labeled A and B. A growth solution was prepared in each of the two vials. First, both vials with 10 mL $H_2O$ contained 1325 μL CTAC (25%), 500 μL of 5 mM HAuCl$_4$ solution, 10 μL of 0.01 M NaBr, and 90 μL of 0.04 M ascorbic acid. Next, 25 μL of the seed solution was added to the solution in vial A for 10 min. Then 25 μL of the solution in vial A was transferred to vial B and stirred for another 15 min. Finally, Au nanocubes were collected and stored in $H_2O$.

### Structural characterization in XRD
The crystal structures were determined by synchrotron X-ray diffraction analysis (the incident X-ray wavelength of 0.7749 Å) using a large Debye−Scherrer camera in BL01C2 at the Taiwan Light Source (TLS) of the National Synchrotron Radiation Research Center in which the electron storage ring was operating at 1.5 GeV and 360 mA under top-up injection. Powder X-ray diffraction patterns of samples have been collected by transmission type. The powder samples are sealed in two Scotch tapes in the glove box of Ar atmospheres for preventing oxidization from the air. Two-dimensional powder X-ray diffraction patterns were recorded by the MAR345 imaging plate. One-dimensional PXRD profiles were integrated from the 2D patterns by GSAS-II software[29]. The XRD patterns were calibrated by the CeO$_2$ standard and altered to that with a wavelength of 1.5406 Å for easy comperes. Rietveld refinement was applied to analyze crystalline phases by using GSAS software[30].

### X-ray absorption spectroscopy analysis
X-ray absorption spectroscopy including X-ray absorption near edge spectra (XANES) and extended X-ray absorption fine structure (EXAFS) at Cu K-edge and Au $L_3$-edge were collected in transmission type at BL01C1 (TLS) and TPS 44A at Taiwan Photon Source (TPS). The scan range was kept in an energy range of 8700–9800 eV for Cu K-edge and 11,719–12,719 eV for Au $L_3$-edge for transmission type. Due to the low Au loading, the Au $L_3$-edge of Au$_{0.02}$Cu$_{0.98}$ was measured in fluorescence mode at TPS 44A. Since Cu has a fluorescence line near that of Au $L$ L$\alpha$ and there is a large loading of Cu, an energy-resolved fluorescence spectrometer using seven-element silicon drift detectors (SDD) was required to detect the weak Au fluorescence signal without saturating the detector with fluorescence form copper[25]. Subtracting the baseline of pre-edge and normalizing that of post-edge obtained the spectra using Athena software. EXAFS analysis was conducted using Fourier transform on $k^3$-weighted EXAFS oscillations to evaluate the contribution of each shell to the Fourier transform peak and fitted by using Artemis software[31].

### Computational details
All periodic DFT calculations in this study were performed with the generalized gradient approximation (GGA) of Perdew−Burke−Ernzerhof (PBE)[32] change-correlation functional employing Vienna ab initio simulation program (VASP)[33–36]. The projector augmented wave (PAW)

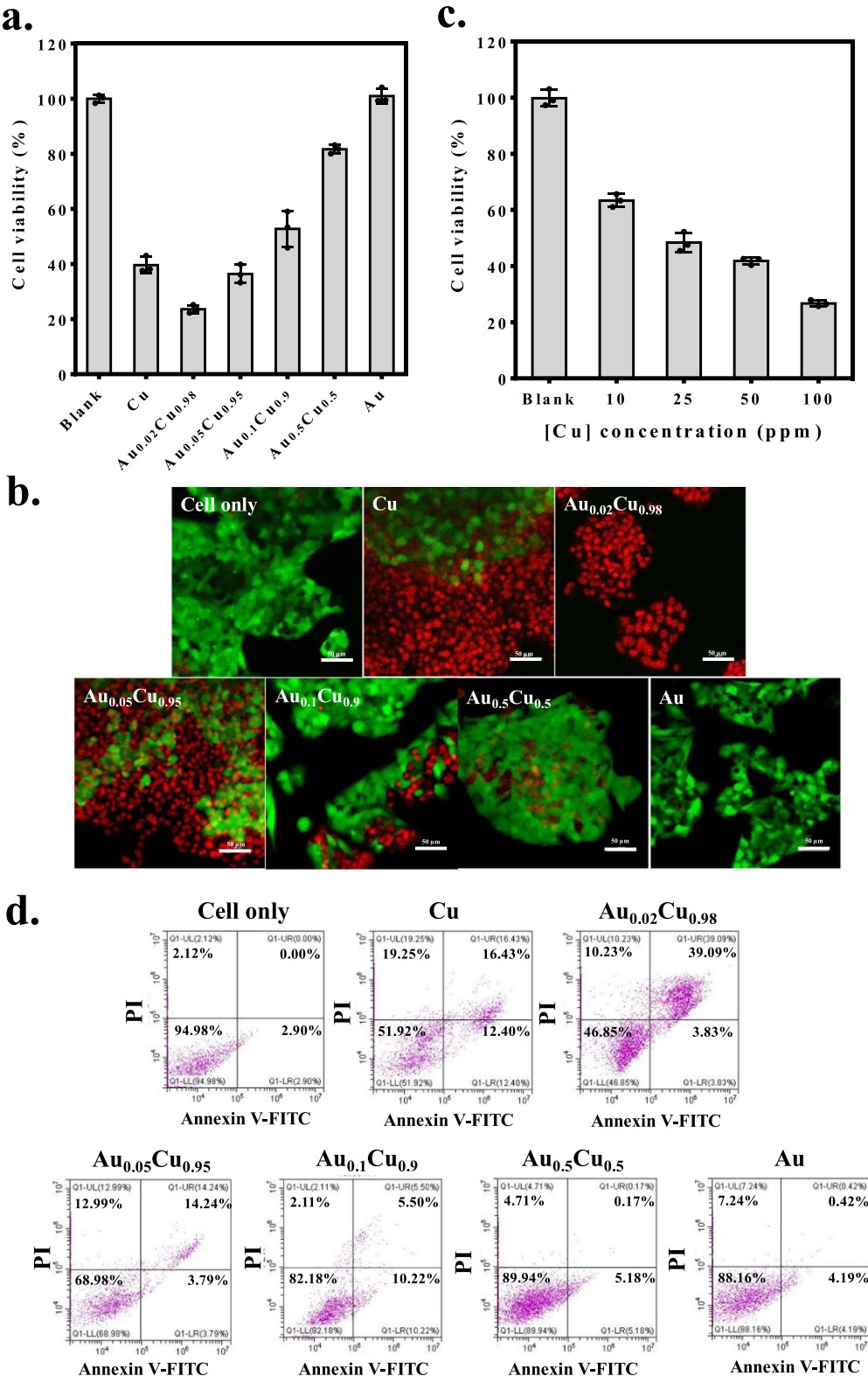

**Fig. 5 | In vitro studies of Cu@SA, Au@SA, and Au/Cu0@SA nanocubes. a** The cytotoxicity study of Cu@SA, Au@SA, and Au/Cu⁰@SAnanocubes with HepG2-Red-FLuc liver cancer cells by using MTT assay through 100 ppm in Cu concentration. **b** Live (green color) and dead cells (red color) stained with fluorescent green dye (Calein-AM) and red dye (propidium iodide), respectively, for cancer cells treated with Cu@SA, Au@SA, and Au/Cu⁰@SA nanocubes (one representative data was shown from three independently repeated experiments). **c** The profiles of cells viability with different Au$_{0.02}$Cu$_{0.98}$@SA concentrations for 24 h incubation at 37 °C. **d** Flow cytometry analysis of HepG2-Red-FLuc cancer cells treated with Cu@SA, Au@SA, and Au/Cu⁰@SA nanocubes. All data were obtained in triplicate (*n* = 3, the error bars represented mean ± SD). Source data are provided as a Source Data file.

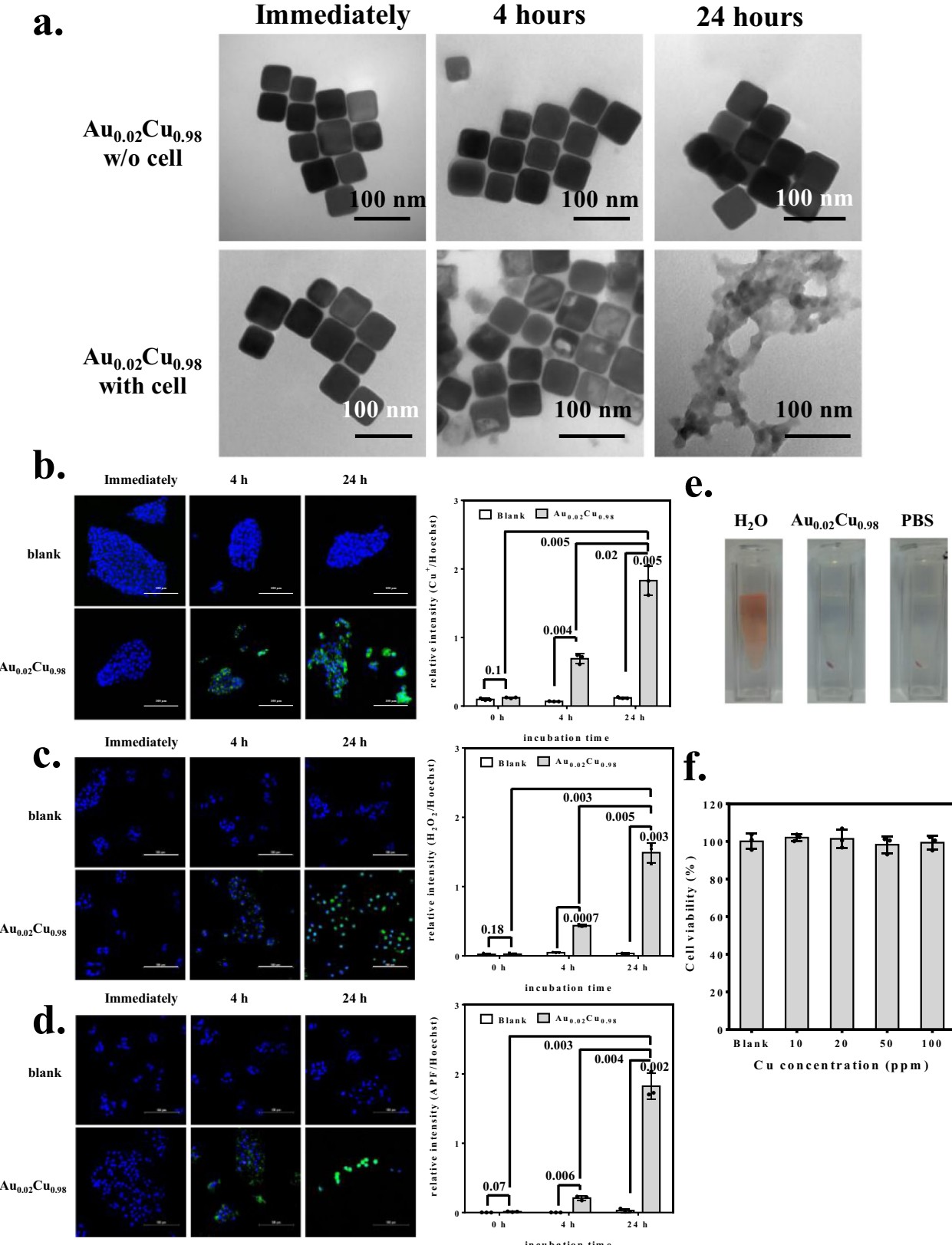

method[37–39] is applied to describe the electron core interactions. The Kohn–Sham orbitals are expanded in a plane-wave basis set with a kinetic energy cutoff of 400 eV. Spin polarization calculation was also involved in the calculation. To carry out the dispersion correction, we performed DFT-D3 functional in our calculation[40]. The convergence threshold was set to be $10^{-5}$ eV for the total electronic energy in the self-

consistent loop. The atomic positions were relaxed using either the conjugate gradient algorithm until the $x$, $y$, and $z$-components of unconstrained atomic force were smaller than $2 \times 10^{-2}$ eV/Å.

All $Au/Cu^0$ nanocubes systems were modeled by the FCC cubic unit cell, including pure Cu, $Au_{0.02}Cu_{0.98}$, $Au_{0.5}Cu_{0.5}$, and pure Au. The (100) slab model was considered for these $Au/Cu^0$ nanocubes by using

**Fig. 6 | In vitro studies of Cu@SA, Au@SA, and Au/Cu0@SA nanocubes. a** The morphology of $Au_{0.02}Cu_{0.98}$@SA nanocubes treated with or without HepG2-Red-FLuc cancer cells as a function of time. **b** $Cu^+$ release stained by CopperGreen dye showing green color as a function of time and the corresponding quantitative analysis. **c** $H_2O_2$ generation stained by hydrogen peroxide assay kit showing green color as a function of time and the corresponding quantitative analysis. **d** •OH generation stained by APF dye showing green color as a function of time and the corresponding quantitative analysis. **e** Analysis of hemolysis in blood containing 2% red blood cells from $Au_{0.02}Cu_{0.98}$@SA nanocubes. Negative and positive controls were conducted by immersing red blood cells in PBS and water, respectively. **f** Cytotoxicity analysis of vascular endothelial cells treated with $Au_{0.02}Cu_{0.98}$@SA nanocubes. The scale bar is 100 μm in all fluorescence images. All data were obtained in triplicate ($n = 3$, the error bars represented mean ± SD, and $p$-values were calculated by one-way ANOVA). Source data are provided as a Source Data file (one representative data was shown from three independently repeated experiments).

$(3 \times 3)$ lateral supercells with a vacuum space of 15 Å between the slab and its periodic replicas. The atomic layers in the slab in this study were constructed with six atomic layers models where the bottom three layers are fully fixed, as shown in Supplementary Fig. S23. The Brillouin zone was sampled with Monkhorst–Pack mesh[41] to be $(10 \times 10 \times 10)$ and $(5 \times 5 \times 1)$ for the Au/$Cu^0$ nanocubes unit cell and supercell, respectively. In addition, the Climbing Image Nudged Elastic Band (CI-NEB) method[42–44] was applied for finding transition states and the minimum energy path of all reactions. The adsorption energy ($E_{ads}$) of the species with the surfaces is defined by the formula: $E_{ads} = E_{mole./sur.} - E_{mole.} - E_{sur.}$, where $E_{sur.}$ is the total energy of the Au/$Cu^0$ nanocubes, $E_{mole.}$ is the total energy of a molecule in gas-phase, and $E_{mole./sur.}$ is the total energy of the Au/$Cu^0$ nanocubes together with the adsorbate. Furthermore, to realistically simulate the effect of the catalytic environment in the solution, we considered the implicit solvent model by using vaspsol[45,46]. The implicit solvent effect of water was adopted in this work, where the corresponding dielectric constant of water is 78.3553.

To verify our computational settings, we have compared our results with some references regarding $O_2$ adsorption on the pure Cu and Au surface, including both experimental and theoretical reports. Experimentally, Cruickshank et al. reported that both O atom and $OH^-$ could be chemisorbed on the 4-fold hollow site on the Cu(100) surface[47]. Compared with our calculated results, the adsorption of the $O_2$ molecule on the 4-fold hollow site via side-on configuration is also the most stable adsorption structure of $O_2$ on the Cu(100) surface. Our results also reveal that the most stable adsorption structures of the OH group and O atom are all on the 4-fold hollow site on the Cu(100) surface. Thus, the trends of our computational results are in good agreement with the experimental observations. Theoretically, Gómez et al. have calculated that the most stable adsorption geometries of $O_2$, O, OH, and $H_2O$ species on the Cu(100) surface are the hollow, hollow, hollow, and top sites, respectively[48]. These results also have similar trends to our results.

For pure Au surface, Kim and Gewirth have observed the oxygen molecular adsorption on Au(100) surface via the SERS study[49]. In addition, some theoretical studies have reported that the most stable adsorption site on Au(100) surface for oxygen molecules is at the hollow site[49,50]. A recent DFT study by Oguz et al. showed that the adsorption energy of $O_2$ on the hollow site at Au(100) surface with PBE functional and DFT-D3 correction is −0.36 eV[50], which is similar to our results. Hence, according to these comparisons, our computational setting is reliable.

## SA modified Au/$Cu^0$ nanocubes
Briefly, Au/$Cu^0$ nanocubes solution (1000 ppm) in 0.5 ml ethanol was mixed with 0.5% SA and sonicated for 5 min. Then, 1 mL of water was added to the mixture and sonicated for another 10 min, followed by centrifugation, and washing with water to remove any excess SA. The obtained pellet was suspended and diluted based on the requirements of the experiments.

## Evaluation of $H_2O_2$ generation
For $H_2O_2$ detection, a colorimetric analysis in the presence of $H_2O_2$, $KMnO_4$ was dissolved in an aqueous solution containing $H_2SO_4$. The mixture was treated with Au/$Cu^0$ nanocubes for 10 min. Subsequently,

the UV−vis spectra were measured from 400 to 650 nm. The $H_2O_2$ quantitative of Au/$Cu^0$ nanocubes was measured by using a probe hydrogen peroxide assay kit. The calibration curve was prepared in serial dilutions of 300 μM $H_2O_2$. The 20 ppm of Au/$Cu^0$ nanocubes were mixed with a hydrogen peroxide assay kit for 10 min. Then, the fluorescence was measured at the emission wavelength of 510 nm (excitation wavelength of 490 nm) using a spectrofluorometer. The hydrogen peroxide assay kit solution alone in PBS was also carried out as a control group.

## Evaluation of •OH generation capability
The •OH generation of Au/$Cu^0$ nanocubes reacted with different pH of PBS were measured by using a probe terephthalic acid. The 20 ppm of Au/$Cu^0$ nanocubes at various pH were mixed with terephthalic acid (0.1 M) for 10 min. Then, the terephthalic acid fluorescence was measured at an emission wavelength of 425 nm (excitation wavelength of 315 nm) using a spectrofluorometer. The terephthalic acid solution alone in PBS was also carried out as a control group.

## Detection of •OH by ESR
The generation of •OH was evaluated by an ESR spectrometer using a DMPO spin-trapping adduct. During the experiments, the solutions included 100 mM DMPO and nanocubes. All mixtures were dispersed in PBS (0.1 M). The solutions were then aspirated into quartz capillaries for ESR analysis.

## Stability analysis
For the stability examination, the $Au_{0.02}Cu_{0.98}$ nanocubes and $Au_{0.02}Cu_{0.98}$@SA nanocubes were individually dispersed in phosphate-buffered saline (PBS) (pH 7), PBS (pH 5), and deionized water in the Eppendorf tubes and incubated at 37 °C and observed for 7 days. For the structural stability of nanocubes at days 0, 1, 3, 5, and 7, the solutions were centrifuged, washed thrice with deionized water, and observed under TEM.

## Cell culture
HepG2-Red-FLuc (Human hepatocellular carcinoma cell line) cells were maintained in MEM supplemented with 10% FBS and the antibiotics penicillin/streptomycin All cells were maintained at 37 °C in a humidified atmosphere containing 5% $CO_2$. HUV-EC-C cells (endothelial cell line) were cultured in F-12k containing EGCS (0.03 mg/mL), heparin (0.1 mg/mL), and fetal bovine serum (FBS, 10%) in the incubator at 37 °C and 5% $CO_2$. The anoxic condition was operated on by maintaining gas concentration at 2% $O_2$. HepG2-Red-FLuc cells were obtained from PerkinElmer (Product No. BW134280).

## In vitro cytotoxicity test
The standard methyl thiazolyltetrazolium (MTT) assay was utilized to assess the toxicity of Au/$Cu^0$ nanocubes in the HepG2-Red-FLuc hepatocellular carcinoma cell line. The cells were cultured in 96-well plates ($1 \times 10^5$ cells/well) for 24 h in complete media prior to treatment with various Cu concentrations in nanocubes at 37 °C under 5% $CO_2$ for 24 h. After washing the cells with PBS buffer, the fresh media containing MTT reagent (0.5 mg mL$^{-1}$) was added and incubated for another 4 h. Then, the medium was then substituted with DMSO to

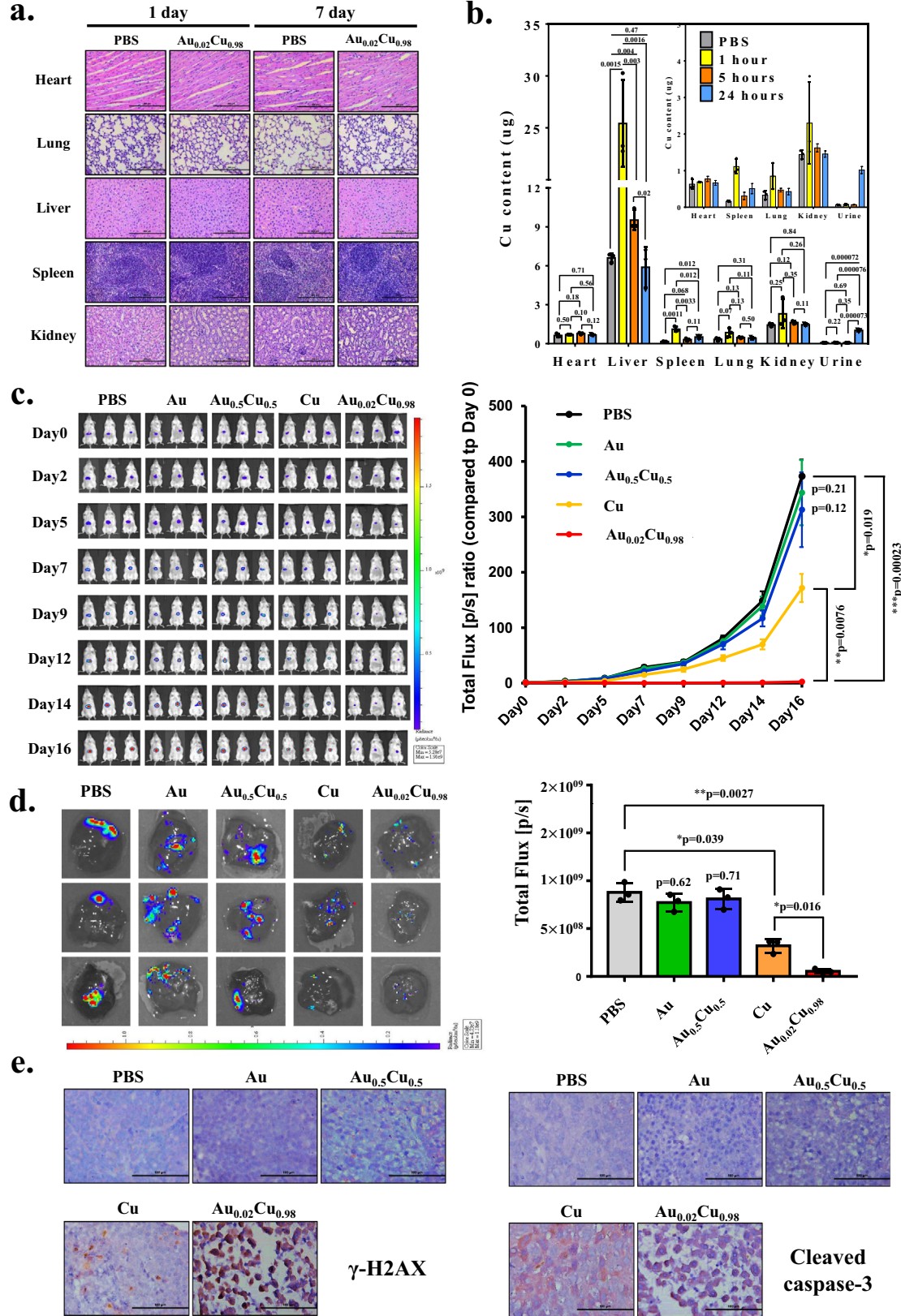

dissolve the resulting formazan. The absorbance was measured at a wavelength of 540 nm by using an ELISA reader.

## Live and dead cells assay

The dyes of propidium iodide (PI) and Calein-AM were used to stain the dead and live cells, respectively. The HepG2-Red-FLuc cancer cells were seeded in 96-well plates (8000 cells/per well) and incubated for 24 h, followed by the treatments of medium alone (blank) or 100 ppm of Au/Cu[0] nanocubes for another 24 h. After treatments, the cells were gently rinsed twice and further stained by PI and Calein-AM following the standard process. The distribution of dead and live cells was observed by laser scanning confocal microscope.

**Fig. 7 | In vivo anti-tumor activity of the mice with HepG2-Red-FLuc orthotopic tumors. a** Histology morphology of each organ with PBS and $Au_{0.02}Cu_{0.98}$@SA nanocubes on post-injection day 7 was observed by H&E staining (scale bar, 200 μm). **b** The biodistribution was determined by Cu concentration collected from $Au_{0.02}Cu_{0.98}$@SA nanocubes through intravenous injection (the inset showing accumulation without liver, $n = 3$). **c** Antitumor efficacy of different nanocubes (Au@SA, $Au_{0.5}Cu_{0.5}$@SA, Cu@SA, and $Au_{0.02}Cu_{0.98}$@SA) in HepG2-Red-FLuc orthotopic liver tumor mice ($n = 3$). Tumor growth was monitored by the IVIS

system. **d** The IVIS bioluminescence of livers with hepatocellular carcinoma in each treatment group after mice were sacrificed ($n = 3$). **e** The expression of phospho-histone H2A.X (Ser139) and cleaved caspase-3 (Asp175) within hepatocellular carcinoma from each treatment group mice by IHC staining (scale bar, 100 μm). Experiments of H&E staining and IHC staining were repeated at least three times independently with similar tendencies and the result from a representative experiment was shown. The error bars represented mean ± SEM (**b**–**d**). The $p$-value was calculated by one-way ANOVA.

## Flow cytometry assay

HepG2-Red-FLuc hepatocellular carcinoma cells were seeded in a 6-cm culture dish with a population of $5 \times 10^5$ cells and incubated overnight. Cells were then treated with $Au/Cu^0$ nanocubes at 100 ppm. Control cells (culture medium only as negative control and 2 μM Thapsigargin as positive control) were also included in this experiment. After 24 h, cells were washed twice with PBS (phosphate buffer saline) and were later detached by trypsinisation. Then, the cells were harvested and washed with PBS. Cells were then re-suspended in 500 μL of 1X annexin V binding buffer. Next, 10 μL of annexin-V(FITC) and 10 μL of propidium iodide were added to cells. Cells were incubated at room temperature for 15 min and analyzed by flow cytometry. Cell populations were initially gated using forward scatter and side scatter plots of the cell-only samples. The gate was set to remove dead cells and aggregates of cells and then applied to all samples (a figure exemplifying the gating strategy is provided in Supplementary Fig. 24).

## In vitro Cu⁺ detection

The HepG2-Red-FLuc cancer cells were seeded in eight-well plates (10,000 cells/per well) and incubated for 24 h, followed by the treatments of CopperGreen dyes with the $Au_{0.02}Cu_{0.98}$@SA nanocubes for another 4 and 24 h. The cells were treated by the medium as the control group. And then, the cells were gently rinsed twice for further observation by laser scanning confocal microscope

## In vitro H₂O₂ detection

The HepG2-Red-FLuc cancer cells were seeded in eight-well plates (10,000 cells/per well) and incubated for 24 h, followed by the treatments of Hydrogen Peroxide Assay Kit with the $Au_{0.02}Cu_{0.98}$@SA nanocubes for another 4 and 24 h. The cells were treated by the medium as the control group. And then, the cells were gently rinsed twice for further observation by laser scanning confocal microscope

## In vitro •OH detection

The HepG2-Red-FLuc cancer cells were seeded in eight-well plates (10000 cells/per well) and incubated for 24 h, followed by the treatments of APF dyes (5 mM) with the $Au_{0.02}Cu_{0.98}$@SA nanocubes for another 4 and 24 h. The cells were treated by the medium as the control group. And then, the cells were gently rinsed twice for further observation by laser scanning confocal microscope

## Hemolysis analysis

The 2% red blood cells were prepared in deionized water (positive control group), PBS (negative control group), and PBS containing with 20 ppm $Au_{0.02}Cu_{0.98}$@SA nanocubes. These solutions were stood at dark for 1 h. And then, the solutions were centrifugated at 6200×g for 5 min to evaluate the hemolysis condition.

## Biosafety in vivo

Animal care was provided in accordance with the Laboratory Animal Welfare Act and the Guide for the Care and Use of Laboratory Animals and approved by the Institutional Animal Care and Use Committee of National Cheng Kung University (NCKU). All animal treatments and surgical procedures were performed in accordance with the guidelines of NCKU Laboratory Animal Center (IACUC No. 111177). The experimental mice were housed in cages (three to five mice in each cage) at

22–23 °C and 55 ± 10% humidity with 13 h/11 h light/dark cycle. C57BL/6 mice (6–8 weeks of age, female) were treated with 100 μL sterilized PBS or 100 μL 600 ppm $Au_{0.02}Cu_{0.98}$@SA (in sterilized PBS) through intravenous administration. The body weight of each group was recorded daily. After 7 days post-treatment, the blood was collected for biochemical analysis and normal organs (heart, lung, spleen, liver, and kidney) were gathered for H&E staining.

## Hematoxylin and eosin (H&E) staining

The samples of tumors and other normal organs (heart, lung, spleen, liver, and kidney) were paraffin-embedded and then sliced into 5 μM thickness. The sections were deparaffinized, rehydrated, washed in PBS, and stained with hematoxylin solution for 3 min. After washing in tap water, the sections were stained with eosin solution for 1 min. Finally, the sections were immersed in ethanol and xylene and mounted for evaluation. The sections were observed under a microscope Olympus BX51 (Olympus, Japan), and three different fields for each group were taken.

## Biochemical analysis

The mice's blood was obtained from the heart, and then heparin sodium was added immediately. The collected blood samples were centrifuged at $100 \times g$ for 10 min to obtain the serum. The serum samples were used for the blood biochemistry analysis for alanine aminotransferase (ALT), alkaline phosphatase (ALP), aspartate aminotransferase (AST), total bilirubin (T-Bil), blood urea nitrogen (BUN), creatine (CREA), and uric acid (UA) expression by a FUJI DRI-CHEM 4000i (FUJIFILM).

## Biodistribution in vivo

Female SCID mice aged 4–7 weeks were obtained from the laboratory animal center, National Cheng Kung University, Taiwan. The $Au_{0.02}Cu_{0.98}$@SA nanocubes (600 ppm in Cu concentration, 100 μL) were intravenously injected into the SCID healthy mice ($n = 3$ per group). The control group with only PBS was also examined. The major organs of the mice (heart, liver, spleen, lungs, and kidneys) and urine were harvested and weighed and then the Cu content was analyzed by ICP-AES.

## Establishment of HepG2-Red-FLuc orthotopic hepatocellular carcinoma model

NOD-SCID mice (6–8 weeks, female) were anesthetized using an intraperitoneal injection of Zoletil 100 (Virbac) and put in a face-up position. Then, $2 \times 10^6$ HepG2-Red-Fluc cells in a solution containing 10 μL of PBS and 10 μL of Basement Membrane Matrix (BD) were surgically implanted into either right or left lobe of the liver using BD Insulin Syringes 30 G 3/10cc (BD). The wound was sutured using CT204 Chromic Catgut (20 mm, 75 cm, UNIK SURGICAL SUTURES MFG. CO.) as well as NC193 Monofilament Nylon (19 mm, 45 cm, UNIK SURGICAL SUTURES MFG. CO.) and the mice were allowed to rest until they fully recover. The tumor growth of HepG2-Red-FLuc hepatocellular carcinoma cells were monitored by the IVIS imaging system (Caliper Life Sciences). The maximal tumor burden permitted by Institutional Animal Care and Use Committee of NCKU was the weight of tumor should not exceed 10% of body weight and without ascites formation. All the experimental mice with

orthotopic hepatocellular carcinoma were sacrifice before reaching the above standard.

## IVIS imaging system and quantification

Mice were anesthetized with a mixture of oxygen and isoflurane and intraperitoneally injected with 100 μL of D-luciferin (catalog # 122796, Caliper Life Sciences). Later at 10 min, mice underwent imaging with the Xenogen IVISR Spectrum Noninvasive Quantitative Molecular Imaging System (Caliper Life Sciences) and analyzed with Living Image 4.7.3 (PerkinElmer, USA). After mice sacrifice, the liver organ was then collected and subjected to IVIS detection ex vivo.

## Immunohistochemistry (IHC) staining

The tumor samples were paraffin-embedded and then sliced into 5 μm thicknesses. The sections were deparaffinized, rehydrated, incubated with phospho-histone H2A.X (Ser139) antibody (1:400 dilution, #9718, Cell Signaling Technology) or cleaved caspase-3 (Asp175) antibody (#9661, Cell Signaling Technology), and then stained by an ABC peroxidase standard staining kit (Thermo Fisher Scientific) containing biotinylated affinity-purified goat anti-rabbit IgG (1:1000 dilution, 32054, Thermo Fisher Scientific) and a DAB peroxidase (HRP) substrate kit (Vector Laboratories) according to the manufacturer's protocol. Finally, the sections were observed under a microscope Olympus BX51 (Olympus) with three different fields taken for each group.

## Reporting summary

Further information on research design is available in the Nature Portfolio Reporting Summary linked to this article.

## Data availability

All data generated that support the findings of this study are present in the article and supplementary information. Besides. Source data are provided with this paper.

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

## Acknowledgements

C.-S.Y. appreciates the financial support provided by the Ministry of Science and Technology, Taiwan (109-2113-M-006-011-MY3). W.-P.S. appreciates the financial support provided by the Ministry of Science and Technology or National Science and Technology Council, Taiwan (109-2314-B-006-084-MY3; 111-2321-B-006-011; 111-2314-B-006-004). Y.-H.C. appreciate the financial support provided by the Ministry of Science and Technology, Taiwan (MOST 111-2113-M-035-003-MY2). H.-S.S. appreciates the TLS 01C1 and TPS44A beamline stuffs Dr. Ting-Shan Chan, Dr. Tai-Sing Wu, Dr. Jeng-Lung Chen for their helps in XAS data collection. C.-H.Y. appreciates the financial support provided by the Ministry of Science and Technology, Taiwan (109-2113-M-035-003-MY3).

## Author contributions

C.-S.Y. conceived the research. C.-S.Y., Y.-H.C., H.-S.S., W.-P.S. and L.-C.W. designed experiments, interpreted data, and wrote manuscript. L.-C.W. designed experiments, materials preparation and characterization and cells studies. H.-S.S., P.-Y.C. and C.-W.P. conducted Synchrotron powder X-ray diffraction, XANES and EXAFS measurements. C.-H.Y. performed Density Functional Theory (DFT) calculation studies. W.-P.S., L.-C.C. and Y.-F.L. conducted animal studies. W.-Q.C. prepared materials and characterization and cells imaging.

## Competing interests

The authors declare no competing interests.
