## [Peer Review File · Nature Communications]

Atomically dispersed golds on degradable zero-valent copper nanocubes augment oxygen driven Fenton-like reaction for effective orthotopic tumor therapyReviewers' Comments:

Reviewer #1:

Remarks to the Author:

Wang et. al. employed a galvanic replacement approach to create atomically dispersed Au on degradable zero-valent Cu nanotubes. AuxCuy enhanced •OH generation following $O_2 \rightarrow H_2O_2 \rightarrow \bullet OH$, finally killing tumor cells. The idea is very interesting. However, the method for material synthesis is not convincing, and some of the data can not support the conclusions. I am sorry that I cannot be more positive on this occasion but hope that the comments are helpful for your study.

1. Galvanic replacement approach has been widely used to synthesize heterostructures or framework nanostructures by using relatively active metal nanoparticles as seeds. The authors claim that Au single-atom-doped Cu nanoparticles were prepared using the same synthesis approach, however, they do not explain why the same method can form single-atom in their manuscript but not Cu-Au heterojunctions. In principle, when the Au^{3+} in the solution is captured by Cu nanocubes and reduced to Au atoms, the subsequently displaced Au atoms are more inclined to grow and attach to the existing Au sites, rather Cu surface, because the formation energy barrier here is lower. Therefore, the galvanic replacement method in solution is theoretically unfavorable for the formation of single atoms.

2. There is also no conclusive evidence in the characterization results to prove the existence of Au single atoms in Cu nanotubes. On the contrary, multiple characterization results indicate that Au exists in the form of crystals rather than a single atomic dispersion on the surface of Cu. The evidence is as follows: 1). The diffraction peaks of Au crystals appear in XRD patterns (Figure 1h) , while Au single atoms do not have crystal diffraction peaks, and the Au grains size calculated from the diffraction peaks are 2.55 nm for $Au_{0.02}Cu_{0.98}$ (Supplementary Table 1). 2). The EXAFS spectrum also cannot support the existence of Au single atoms, because if Au nanoparticles were grown on the surface of Cu, Au-Cu bonds will also appear in the spectrum, and the appearance of Au-Au bonds at 2.82Å is more inclined to corroborate Au in the form of nanoparticles rather than single atoms.

3. The authors take Figure 1.1r as direct evidence for the existence of Au single atoms, but this figure is unconvincing. From the XRD patterns, it is known that Cu has good crystallinity. However, HADDF-STEM, which can distinguish Au single atoms, does not show the atomic image or crystal lattice of Cu, which is incomprehensible.

4. The detection method for H_2O_2 production is not specific and cannot adequately prove that H_2O_2 is produced. Quantitative detection of H_2O_2 production rate and yield is required, which is critical to assess its therapeutic capacity.

5. The biological function of AuxCuy is weak. As shown in Fig5a, only a 10% tumor cell killing effect was increased in $Au_{0.02}Cu_{0.98}$ group, when compared with the Cu group. Furthermore, the function is decreased along with the increase of Au.

6. AuxCuy enhanced •OH generation following $O_2 \rightarrow H_2O_2 \rightarrow \bullet OH$, finally killing tumor cells. Therefore, it is better to perform in vitro or in vivo biological experiments with or without O_2 to further support the conclusion.

7. All the figures are vague, especially in Fig5 and Fig.6. The graphs are too tiny and very hard to read. The pictures of immune staining are very blurry. Please rearrange.

Reviewer #2:

Remarks to the Author:

The authors employ a galvanic replacement approach to create atomically dispersed Au on degradable zero-valent Cu nanocubes. It is found that $Au_{0.02}Cu_{0.98}$ composition reveals the enhanced •OH

generation following $O_2 \rightarrow H_2O_2 \rightarrow \bullet OH$, which favors the excretory system through urinary metabolism when applied in tumoral treatments. The advantage of $Au_{0.02}Cu_{0.98}$ compared with other compositions have been discussed based on density functional theory calculations. For the theoretical calculations, a major revision should be made before the possible acceptance for publication.

1. For the slab model, it is not clear how many atomic layers in the slab has been used. The atomic model should be presented in the Supporting Information.
2. On page 8, it is stated that the surfaces were pre-covered with hydrogen atoms. However, the authors did not clarify and justify the source of the hydrogen atoms. Where does the hydrogen atoms come from? Especially, what is the rule for the distribution of the hydrogen atoms? I think in the solution the hydrogen atom may mainly result from water molecule. So, the production of hydrogen should also be investigated theoretically.
3. The adsorption of hydrogen atoms should also be investigated. Presently, it seems to me that the investigation on hydrogen adsorption is rather not careful.
4. How does the preadsorbed hydrogen affect the adsorption of O_2 ? Will different H adsorption distribution affect the reaction thermodynamics and kinetics?
5. The most stable adsorption configuration is relevant to the subsequent hydrogenation reaction. It is noted that the authors only present one configuration of O_2 adsorption for each substrate. It is suggested that the authors put all the possible adsorption configurations and adsorption energies in the Supporting Information for references.
6. The authors used VASP calculating reaction energies. We know that VASP cannot calculate charged particle due to the energy divergency. Thus, it is needed to clarify the calculation of the reaction energy for H_2O_2 to $OH^- + OH$.
7. Due to that the reaction occurs in the solution, the solvation effect should be considered for the calculation.
8. On page 9, the reason for Au carrying negative charge should result from the difference between their electronegativity rather work function.
9. There should be reference comparison for O_2 adsorption on the pure Cu and Au surface to verify the computational settings.

REVIEWER COMMENTS

Reviewer #1 (Remarks to the Author):

Wang et. al. employed a galvanic replacement approach to create atomically dispersed Au on degradable zero-valent Cu nanotubes. Au_xCu_y enhanced •OH generation following O₂→H₂O₂→•OH, finally killing tumor cells. The idea is very interesting. However, the method for material synthesis is not convincing, and some of the data can not support the conclusions. I am sorry that I cannot be more positive on this occasion but hope that the comments are helpful for your study.

1. Galvanic replacement approach has been widely used to synthesize heterostructures or framework nanostructures by using relatively active metal nanoparticles as seeds. The authors claim that Au single-atom-doped Cu nanoparticles were prepared using the same synthesis approach, however, they do not explain why the same method can form single-atom in their manuscript but not Cu-Au heterojunctions. In principle, when the Au³⁺ in the solution is captured by Cu nanocubes and reduced to Au atoms, the subsequently displaced Au atoms are more inclined to grow and attach to the existing Au sites, rather Cu surface, because the formation energy barrier here is lower. Therefore, the galvanic replacement method in solution is theoretically unfavorable for the formation of single atoms.

Response: We thank the reviewer for the constructive comments. To reply the reviewer's inquiry, we have further designed and demonstrated a series of Au_xCu_y nanoparticles in our studies by tuning the amount of HAuCl₄ (as seen in the following **Table R1.**) through the galvanic replacement reaction. The fixed concentration of Cu nanocubes (10⁴ ppm, 100 μL) was used for galvanic replacement reaction accompanied with 50 mM HAuCl₄ in the volumes of 5, 10, 20, 40, 60, 80, and 100 μL, respectively. Consequently, the volume of 100 μL HAuCl₄ could completely replace Cu nanocubes forming Au nanoparticles according to their TEM images and XRD pattern (**See Supplementary Fig. 5**). On the other hand, the alloyed Au_{0.75}Cu_{0.25} nanocages and Au_{0.8}Cu_{0.2} nanoframes occurred during the volumes of HAuCl₄ decreased to 60 and 80 μL, respectively. Both (Au_{0.75}Cu_{0.25} and Au_{0.8}Cu_{0.2}) characteristics including the TEM images, XRD patterns, and HR-TEM images were shown in **Supplementary Fig. 3 and Fig. 4**, individually. The alloyed nanocages and nanoframes could be obviously observed by TEM images and the

corresponding XRD patterns were assigned to fcc crystal structure individually followed JCPDS cards 01-071-5023 and JCPDS cards 01-072-5241. In addition, the HR-TEM images and their related electron diffractions indicate the arrangements of (200) and (110). The relevant descriptions have been added in p.4 and 5 in manuscript. When the volume of H₂AuCl₄ was reduced to 5 μ L resulting in the single-atom Au on Cu nanocubes rather than Cu-Au heterojunctions.

In addition, the previously literature has reported the synthesis of Au₁Cu single-atom alloys through galvanic replacement.¹

Table R1. The synthesis parameters of Au_xCu_y nanoparticles as a function of the amount of H₂AuCl₄.

H ₂ AuCl ₄ (50 mM)	Cu nanocubes (10000 ppm)						
	5 (μ L)	10 (μ L)	20 (μ L)	40 (μ L)	60 (μ L)	80 (μ L)	100 (μ L)
Au _x Cu _y NPs	Au _{0.02} Cu _{0.98}	Au _{0.05} Cu _{0.95}	Au _{0.1} Cu _{0.9}	Au _{0.5} Cu _{0.5}	Au _{0.75} Cu _{0.25}	Au _{0.8} Cu _{0.2}	Au

Reference

1. Zhang Y, Chen X, Wang W, Yin L, & Crittenden JC. Electrocatalytic nitrate reduction to ammonia on defective Au₁Cu (111) single-atom alloys. *Applied Catalysis B: Environmental* 310, 121346 (2022).

2. There is also no conclusive evidence in the characterization results to prove the existence of Au single atoms in Cu nanotubes. On the contrary, multiple characterization results indicate that Au exists in the form of crystals rather than a single atomic dispersion on the surface of Cu. The evidence is as follows: 1). The diffraction peaks of Au crystals appear in XRD patterns (Figure 1h), while Au single atoms do not have crystal diffraction peaks, and the Au grains size calculated from the diffraction peaks are 2.55 nm for Au_{0.02}Cu_{0.98} (Supplementary Table 1). 2). The EXAFS spectrum also cannot support the existence of Au single atoms, because if Au nanoparticles were grown on the surface of Cu, Au-Cu bonds will also appear in the spectrum, and the appearance of Au-Au bonds at 2.82Å is more inclined to corroborate Au in the form of nanoparticles rather than single atoms.

Response: We thank the reviewer for the insightful comments. The most direct evidence for Au single atoms is AC HAADF-STEM, which is clearly shown in **Fig. 1r and Supplementary Fig. 7**.

As for the XRD pattern of Au_{0.02}Cu_{0.98}, there is indeed a diffraction

signal of Au crystal. To observe the XRD signal, there must be crystals of comparable size. It means that there are still some large enough Au crystals in $\text{Au}_{0.02}\text{Cu}_{0.98}$, but the signal is very weak. Therefore, it can only be said that some Au clusters and Au single atoms coexist, which cannot be quantified at present.

As for EXAFS, both crystalline and amorphous Au signals can be seen. There is no problem in measuring low-level trace signals by fluorescence XAS. However, the emission signal of Au $L\alpha$ (9.4 keV) and the absorption energy of Cu K -edge (8.979 keV) are too close and close to the emission of Cu $K\alpha$ (8.0 keV). The Cu content is about 50 times more than that of Au. Therefore, the light emitted by Au will be absorbed by Cu a lot. Therefore, we must use a high energy resolution detector (silicon drift detectors, SDD) to filter out the Cu emission signal. Accordingly, we can still observe Au EXAFS of $\text{Au}_{0.02}\text{Cu}_{0.98}$ and other Au_xCu_y samples containing high Au contents. Therefore, the whole experiments can be fully presented. It is noted that the signal of $\text{Au}_{0.02}\text{Cu}_{0.98}$ is very different from that of pure Au foil, and the EXAFS signal of Au-Cu and its ratio to the signal of Au-Au is also observed, which indicate that there may be the contribution of Au single atom on Cu. Please see 2nd paragraph, p.6. Although the overall EXAFS mixes the signal of Au cluster to Cu and the signal of Au single atom (we cannot obtain the signal of pure Au single atom on Cu at present using EXAFS), it is qualitatively presented from the signal ratio.

3. The authors take Figure 1.1r as direct evidence for the existence of Au single atoms, but this figure is unconvincing. From the XRD patterns, it is known that Cu has good crystallinity. However, HAADF-STEM, which can distinguish Au single atoms, does not show the atomic image or crystal lattice of Cu, which is incomprehensible.

Response: We thank the reviewer for the thoughtful comment. We are sorry to cause this confusion. We have labelled d-spacings of Cu in **Fig. 1r** and also provide the magnified view of **Fig. 1r** in **Supplementary Fig. 7** in AC HAADF-STEM image. The d-spacing of **Fig. 1r** is 0.18 nm corresponding to (200) face of Cu.

4. The detection method for H_2O_2 production is not specific and cannot adequately prove that H_2O_2 is produced. Quantitative detection of H_2O_2 production rate and yield is required, which is critical to assess its therapeutic

capacity.

Response: We thank the reviewer for insightful comments. The quantitative results of H_2O_2 have been added in **Fig. 2d**. We evaluated the generation of H_2O_2 by hydrogen peroxide assay kit in different compositions of Au_xCu_y (Cu, $\text{Au}_{0.02}\text{Cu}_{0.98}$, $\text{Au}_{0.05}\text{Cu}_{0.95}$, $\text{Au}_{0.1}\text{Cu}_{0.9}$, $\text{Au}_{0.5}\text{Cu}_{0.5}$, and Au). The largest H_2O_2 generation was calculated to be 122 μM in $\text{Au}_{0.02}\text{Cu}_{0.98}$ in 10 min of reaction. The amount of H_2O_2 in $\text{Au}_{0.02}\text{Cu}_{0.98}$ is higher than the endogenous H_2O_2 in tumoral microenvironments ($\sim 100 \mu\text{M}$) and can be generated in the presence of O_2 . The relevant statements have been added in p.7 in manuscript.

5. The biological function of Au_xCu_y is weak. As shown in Fig5a, only a 10% tumor cell killing effect was increased in $\text{Au}_{0.02}\text{Cu}_{0.98}$ group, when compared with the Cu group. Furthermore, the function is decreased along with the increase of Au.

Response: The cytotoxicity study of MTT assay (**Fig. 5a**) is based on the total production of formazan which is metabolized by succinate dehydrogenase (SDH) from the living cells. The result from MTT assay is inexact and usually be used to represent the approximate tendency. Therefore, we have also performed the Annexin V and PI double-staining to quantify the dead cell population exactly by using flow cytometry in this study (**Fig. 5d**). The flow cytometry can allow us to analyze the individual cell level among the population, which is more accurate than the data from MTT assay. Importantly, the **Fig. 5d** suggests the percentage in the population of late apoptotic cells in $\text{Au}_{0.02}\text{Cu}_{0.98}$ group is 39.09%, which is significantly higher than that in Cu group (16.43%), indicating the remarkable antitumor cytotoxicity of $\text{Au}_{0.02}\text{Cu}_{0.98}$ to HepG2 hepatocellular carcinoma cells. Moreover, the *in vivo* experiments also demonstrated the best antitumor efficacy in $\text{Au}_{0.02}\text{Cu}_{0.98}$ group with conspicuous tumor regression (**Fig. 7c-e**). Comprehensively, the experimental results supported the Au single atom boosting zero-valent copper to release Cu^+ and Cu^{2+} *via* degradable $\text{Au}_{0.02}\text{Cu}_{0.98}$ resulting in high efficiency of oxygen reduction and Fenton-like reactions.

6. Au_xCu_y enhanced $\cdot\text{OH}$ generation following $\text{O}_2 \rightarrow \text{H}_2\text{O}_2 \rightarrow \cdot\text{OH}$, finally killing tumor cells. Therefore, it is better to perform *in vitro* or *in vivo* biological experiments with or without O_2 to further support the conclusion.

Response: We thank the reviewer for the constructive comments. Because

the HepG2 cells could continuously live in an oxygen supply at 1%~20%,^{1,2} we have further designed and evaluated the *in vitro* experiments in an anoxic condition (2 % in O₂) to investigate H₂O₂ generation and Fenton-like reactions using Au_{0.02}Cu_{0.98}@SA. The HepG2 cells were incubated in 2% O₂ environment for 24 h, and then treated with Au_{0.02}Cu_{0.98}@SA for another 24 h. The very weak fluorescence signals were seen in 2% O₂ condition corresponding to the low amount of H₂O₂ and •OH generation compared with in 20% O₂ condition (**Supplementary Fig. 19**), indicating the support of the O₂→H₂O₂→•OH reactions. The relevant statements have been added in p.12 in manuscript.

Reference

1. Hwang SY, *et al.* Emodin attenuates radioresistance induced by hypoxia in HepG2 cells via the enhancement of PARP1 cleavage and inhibition of JMJD2B. *Oncology reports* 33, 1691-1698 (2015).
2. Jin X, Gong L, Peng Y, Li L, & Liu G. Enhancer-bound Nrf2 licenses HIF-1 α transcription under hypoxia to promote cisplatin resistance in hepatocellular carcinoma cells. *Aging (Albany NY)* 13, 364 (2021).

7. All the figures are vague, especially in Fig5 and Fig.6. The graphs are too tiny and very hard to read. The pictures of immune staining are very blurry. Please rearrange.

Response: We appreciate the reviewer's comments. We have rearranged the original **Fig. 5** and **Fig.6** to **Fig.5-7** and clarify the images.

Reviewer #2 (Remarks to the Author):

The authors employ a galvanic replacement approach to create atomically dispersed Au on degradable zero-valent Cu nanocubes. It is found that Au_{0.02}Cu_{0.98} composition reveals the enhanced •OH generation following O₂→H₂O₂→•OH, which favors the excretory system through urinary metabolism when applied in tumoral treatments. The advantage of Au_{0.02}Cu_{0.98} compared with other compositions have been discussed based on density functional theory calculations. For the theoretical calculations, a major revision should be made before the possible acceptance for publication.

1. For the slab model, it is not clear how many atomic layers in the slab has been used. The atomic model should be presented in the Supporting Information.

Response: Thank the reviewer's thoughtful suggestion. The atomic layers in the slab in this study were constructed with six atomic layers models where the bottom three layers are fully fixed, as shown in the below Figure A1. We have added the atomic model of top and side views in the revised Supporting Information. Please see the computational details in 2nd paragraph, p.16 described as "The atomic layers in the slab in this study were constructed with six atomic layers models where the bottom three layers are fully fixed, as shown in the Supplementary Fig. S23."

Figure A1. Top and side views of the (a) Cu(100), (b) Au_{0.02}Cu_{0.98}(100), (c) Au_{0.5}Cu_{0.5}(100), and (d) Au(100) surfaces, where the T, B, and H represent the top, bridge, and hollow sites. Brown and gold spheres represent Cu and Au atoms, respectively.

2. On page 8, it is stated that the surfaces were pre-covered with hydrogen atoms. However, the authors did not clarify and justify the source of the hydrogen atoms. Where does the hydrogen atoms come from? Especially, what is the rule for the distribution of the hydrogen atoms? I think in the solution the hydrogen atom may mainly result from water molecule. So, the production of hydrogen should also be investigated theoretically.

Response: Thank the reviewer's thoughtful suggestion. We have calculated the reaction mechanisms of hydrogen production from water molecule's dissociation on the Cu(100), Au_{0.02}Cu_{0.98}(100), Au_{0.5}Cu_{0.5}(100), and Au(100) surfaces. The water dehydrogenation reaction to produce the hydrogen atoms is also calculated. As shown in below **Table A1** and **Fig. A2**, the first dehydrogenation barriers of H₂O are similar on pure Cu, Au_{0.02}Cu_{0.98}, and Au_{0.5}Cu_{0.5}, but they are much larger on pure Au. The second dehydrogenation barriers of the OH group on pure Cu and Au_{0.02}Cu_{0.98} are smaller than those on Au_{0.5}Cu_{0.5} and pure Au. It reveals that the hydrogen formation rates on Cu and Au_{0.02}Cu_{0.98} are faster than those on Au_{0.5}Cu_{0.5} and pure Au. Besides, we have also calculated the adsorption energies of H₂O molecule, OH group, H atom, and O atom on

all possible active sites at the Cu(100), Au_{0.02}Cu_{0.98}(100), Au_{0.5}Cu_{0.5}(100), and Au(100) surfaces. The intermediates in the reaction mechanisms have been carried out using the most stable adsorption structures to proceed. All the adsorption energies, activation energies, and reaction energies in the potential energy profiles were also corrected by using the solvent effect. Detailed calculated results have been summarized in the Supplementary Figs. 11-15 and Supplementary Tables 4-8, in the revised Supporting Information. Please also see below Figs. A2-A6 and Tables A1-A5.

The relevant description of hydrogen from water molecule is also added in the revised manuscript as follows (1st paragraph, p.9),
“The water dehydrogenation reaction to produce the hydrogen atoms was also calculated. As shown in Supplementary Table 4, and Supplementary Fig. 11, the first dehydrogenation barriers of H₂O are similar on pure Cu, Au_{0.02}Cu_{0.98}, and Au_{0.5}Cu_{0.5}, but they are much larger on pure Au. The second dehydrogenation barriers of the OH group on pure Cu and Au_{0.02}Cu_{0.98} are smaller than those on Au_{0.5}Cu_{0.5} and pure Au. It reveals that the hydrogen formation rates on Cu and Au_{0.02}Cu_{0.98} are faster than those on Au_{0.5}Cu_{0.5} and pure Au. Besides, we have also calculated the adsorption energies of H₂O molecule, OH group, H atom, and O atom on all possible active sites at the Cu(100), Au_{0.02}Cu_{0.98}(100), Au_{0.5}Cu_{0.5}(100), and Au(100) surfaces, as shown in Supplementary Figs. 11-15 and Supplementary Tables 4-8. The intermediates in the reaction mechanisms were carried out using the most stable adsorption structures to proceed. All the adsorption energies, activation energies, and reaction energies were considered using the solvent effect.”

Table A1. Calculated reaction barriers (E_a in eV) and reaction energies (ΔE in eV) for elementary reactions of H_2O dehydrogenation to H atoms on the Au/Cu⁰ nanocubes. The values outside and inside the parentheses represent with and without considering the solvent effect.

Elementary steps	E_a	ΔE
Cu(100)		
$H_2O_{(a)} \rightarrow H_{(a)} + OH_{(a)}$	1.48 (1.38)	0.10 (-0.06)
$H_{(a)} + OH_{(a)} \rightarrow 2H_{(a)} + O_{(a)}$	1.59 (1.57)	0.52 (0.47)
Au_{0.02}Cu_{0.98}		
$H_2O_{(a)} \rightarrow H_{(a)} + OH_{(a)}$	1.49 (1.41)	0.17 (0.03)
$H_{(a)} + OH_{(a)} \rightarrow 2H_{(a)} + O_{(a)}$	1.62 (1.61)	0.54 (0.50)
Au_{0.5}Cu_{0.5}		
$H_2O_{(a)} \rightarrow H_{(a)} + OH_{(a)}$	1.44 (1.39)	0.83 (0.73)
$H_{(a)} + OH_{(a)} \rightarrow 2H_{(a)} + O_{(a)}$	1.99 (1.99)	1.18 (1.12)
Au(100)		
$H_2O_{(a)} \rightarrow H_{(a)} + OH_{(a)}$	1.88 (1.85)	1.05 (0.93)
$H_{(a)} + OH_{(a)} \rightarrow 2H_{(a)} + O_{(a)}$	2.15 (2.15)	1.93 (1.83)

Figure A2. The calculated potential energy profiles of hydrogen production from H_2O dissociation on the a. pure Cu, b. $\text{Au}_{0.02}\text{Cu}_{0.98}$, c. $\text{Au}_{0.5}\text{Cu}_{0.5}$, and d. pure Au. Brown, gold, red and white spheres represent Cu, Au, O, and H atoms, respectively.

Table A2. The calculated adsorption energy (E_{ads}) of the adsorption of H_2O molecule on all the possible adsorption sites at $\text{Cu}(100)$, $\text{Au}_{0.02}\text{Cu}_{0.98}(100)$, $\text{Au}_{0.5}\text{Cu}_{0.5}(100)$, and $\text{Au}(100)$ surfaces. The adsorption energy values outside and inside the parentheses represent with and without considering the solvent effect.

Surface	Site	E_{ads} (eV)	Surface	Site	E_{ads} (eV)
Cu	T	-0.36 (-0.41)	Au	T	-0.22 (-0.32)
	T_1	-0.39 (-0.46)		T_1	-0.40 (-0.51)
$\text{Au}_{0.02}\text{Cu}_{0.98}$	T_2	-0.35 (-0.41)	$\text{Au}_{0.5}\text{Cu}_{0.5}$	T_2	-0.20 (-0.28)
	T_3	-0.15 (-0.23)			

Figure A3. Optimized adsorption structures of H_2O molecule on (a) T site at $\text{Cu}(100)$ surface; (b) T_1 , (c) T_2 , and (d) T_3 sites at $\text{Au}_{0.02}\text{Cu}_{0.98}(100)$ surface; (e) T_1 , and (f) T_2 sites at $\text{Au}_{0.5}\text{Cu}_{0.5}(100)$ surface; and (g) T site at $\text{Au}(100)$ surface. Brown, gold, red and white spheres represent Cu, Au, O, and H atoms, respectively.

Table A3. The calculated adsorption energy (E_{ads}) of the adsorption of OH group on all the possible adsorption sites at Cu(100), Au_{0.02}Cu_{0.98}(100), Au_{0.5}Cu_{0.5}(100), and Au(100) surfaces. The adsorption energy values outside and inside the parentheses represent with and without considering the solvent effect.

Surface	Site	E_{ads} (eV)	Surface	Site	E_{ads} (eV)
Cu	T(\rightarrow H) ^a	-3.46 (-3.54)	Au	T(\rightarrow B) ^a	-2.59 (-2.67)
	H	-3.46 (-3.54)		H(\rightarrow B) ^a	-2.59 (-2.67)
	B	-3.42 (-3.50)		B	-2.59 (-2.67)
Au _{0.02} Cu _{0.98}	T ₁ (\rightarrow H ₁) ^a	-3.42 (-3.51)	Au _{0.5} Cu _{0.5}	T ₁	-2.33 (-2.40)
	T ₂ (\rightarrow H ₁) ^a	-3.42 (-3.51)		T ₂	-1.65 (-1.68)
	T ₃ (\rightarrow B ₂) ^a	-2.97 (-3.03)		B	-2.94 (-3.03)
	B ₁	-3.37 (-3.45)		H ₁	-2.85 (-2.95)
	B ₂	-2.97 (-3.03)		H ₂	-2.85 (-2.94)
	H ₁	-3.42 (-3.51)			
	H ₂	-3.18 (-3.28)			

^a The left and right parts of the arrow show that the adsorbate is adsorbed on the original active sites before optimization and on the final active sites after optimization, respectively.

Figure A4. Optimized adsorption structures of OH group on (a) H site and (b) B site at Cu(100) surface; (c) H₁, (d) H₂, (e) B₁, and (f) B₂ sites at Au_{0.02}Cu_{0.98}(100) surface; (g) H₁, (h) H₂, (i) T₁, (j) T₂, and (k) B sites at Au_{0.5}Cu_{0.5}(100) surface; and (l) B site at Au(100) surface. Brown, gold, red and white spheres represent Cu, Au, O, and H atoms, respectively.

Table A4. The calculated adsorption energy (E_{ads}) of the adsorption of O atom on all the possible adsorption sites at Cu(100), Au_{0.02}Cu_{0.98}(100), Au_{0.5}Cu_{0.5}(100), and Au(100) surfaces. The adsorption energy values outside and inside the parentheses represent with and without considering the solvent effect.

Surface	Site	E_{ads} (eV)	Surface	Site	E_{ads} (eV)
Cu	T(\rightarrow H) ^a	-5.58 (-5.51)	Au	T(\rightarrow B) ^a	-3.65 (-3.51)
	H	-5.58 (-5.51)		H	-3.54 (-3.45)
	B(\rightarrow H) ^a	-5.58 (-5.51)		B	-3.65 (-3.51)
Au _{0.02} Cu _{0.98}	T ₁ (\rightarrow H ₁) ^a	-5.53 (-5.46)	Au _{0.5} Cu _{0.5}	T ₁	-3.38 (-3.10)
	T ₂ (\rightarrow H ₁) ^a	-5.53 (-5.46)		T ₂	-3.00 (-2.69)
	T ₃ (\rightarrow H ₂) ^a	-5.09 (-5.00)		B	-4.20 (-4.06)
	B ₁ (\rightarrow H ₂) ^a	-5.09 (-5.00)		H ₁	-4.31 (-4.22)
	B ₂ (\rightarrow H ₂) ^a	-5.09 (-5.00)		H ₂	-4.58 (-4.51)
	H ₁	-5.53 (-5.46)			
	H ₂	-5.09 (-5.00)			

^a The left and right parts of the arrow show that the adsorbate is adsorbed on the original active sites before optimization and on the final active sites after optimization, respectively.

Figure A5. Optimized adsorption structures of O atom on (a) H site at Cu(100) surface; (b) H₁, and (c) H₂ sites at Au_{0.02}Cu_{0.98}(100) surface; (d) H₁, (e) H₂, (f) B, (g) T₁, (h) T₂ sites at Au_{0.5}Cu_{0.5}(100) surface; and (i) B and (j) H site at Au(100) surface. Brown, gold, and red spheres represent Cu, Au, and O atoms, respectively.

Table A5. The calculated adsorption energy (E_{ads}) of the adsorption of H atom on all the possible adsorption sites at Cu(100), Au_{0.02}Cu_{0.98}(100), Au_{0.5}Cu_{0.5}(100), and Au(100) surfaces. The adsorption energy values outside and inside the parentheses represent with and without considering the solvent effect.

Surface	Site	E_{ads} (eV)	Surface	Site	E_{ads} (eV)
Cu	T(\rightarrow H) ^a	-3.62 (-3.61)	Au	T	-3.10 (-3.08)
	H	-3.62 (-3.61)		H(\rightarrow B) ^a	-3.39 (-3.37)
	B	-3.51 (-3.49)		B	-3.39 (-3.37)
Au _{0.02} Cu _{0.98}	T ₁ (\rightarrow H ₁) ^a	-3.62 (-3.61)	Au _{0.5} Cu _{0.5}	T ₁ (\rightarrow H ₁) ^a	-3.43 (-3.41)
	T ₂ (\rightarrow H ₂) ^a	-3.47 (-3.46)		T ₂	-3.21 (-3.19)
	T ₃	-3.12 (-3.08)		B	-3.39 (-3.38)
	B ₁	-3.48 (-3.46)		H ₁	-3.43 (-3.41)
	B ₂	-3.40 (-3.38)		H ₂ (\rightarrow B) ^a	-3.39 (-3.38)
	H ₁	-3.62 (-3.61)			
	H ₂	-3.47 (-3.46)			

^a The left and right parts of the arrow show that the adsorbate is adsorbed on the original active sites before optimization and on the final active sites after optimization, respectively.

Figure A6. Optimized adsorption structures of H atom on (a) B and (b) H sites at Cu(100) surface; (c) B₁, (d) B₂, (e) H₁, (f) H₂, and (g) T₃ sites at Au_{0.02}Cu_{0.98}(100) surface; (h) B, (i) H₁, (j) T₂ sites at Au_{0.5}Cu_{0.5}(100) surface; and (k) B and (l) T sites at Au(100) surface. Brown, gold, and white spheres represent Cu, Au, and H atoms, respectively.

3. The adsorption of hydrogen atoms should also be investigated. Presently, it seems to me that the investigation on hydrogen adsorption is rather not careful.
Response: Thank the reviewer's thoughtful suggestion. We have added the adsorption energy of hydrogen atom on all the possible active sites at four different metal surfaces, as shown in above **Table A5** and **Fig. A6**. These results were also added in the revised Supporting information (**Supplementary Table 8** and **Supplementary Fig. 15**). All the adsorption

energies of hydrogen atoms at each metal surfaces were also corrected by using the solvent effect.

4. How does the preadsorbed hydrogen affect the adsorption of O₂?

Response: Thank the reviewer's thoughtful comment. The preadsorbed hydrogen can decrease the adsorption energy of the O₂ molecule. For example, the adsorption of O₂ on the most stable sites of the Cu(100), Au_{0.02}Cu_{0.98}(100), Au_{0.5}Cu_{0.5}(100), and Au(100) surfaces are -2.12 eV, -2.00 eV, -1.16 eV, and -0.42 eV, respectively. After pre-covered the hydrogen atoms, the adsorption energy of O₂ becomes -1.95 eV, -1.83 eV, -1.11 eV, and -0.25 eV on the Cu(100), Au_{0.02}Cu_{0.98}(100), Au_{0.5}Cu_{0.5}(100), and Au(100) surfaces, respectively. Thus, the preadsorbed hydrogen can affect the strength of the adsorption energy for the O₂ molecule.

Will different H adsorption distribution affect the reaction thermodynamics and kinetics?

Response: Thank the reviewer's thoughtful comment. The different H adsorption distribution can affect the reaction thermodynamics and kinetics of O₂ hydrogenation only on the Au_{0.02}Cu_{0.98}(100) surface. As shown in below **Fig. A7 (Fig.4** in manuscript), there are two possible reaction mechanisms of O₂ hydrogenation on the Au_{0.02}Cu_{0.98}(100) surface. The first pathway did not go through *via* the Au atom, and then the two hydrogenation barriers are 0.98 eV and 0.99 eV. On the other hand, the second hydrogenation pathway involves the Au atom, and the two hydrogenation barriers become 0.75 eV and 0.72 eV. The reason is that the Au atom in the Au_{0.02}Cu_{0.98}(100) surface will create a different chemical environment of the copper atoms and thus results in the different reaction barriers for the situation of different H adsorption distributions. For the other three metal surfaces, the chemical environment of either copper or gold is all the same so that the H adsorption distribution will not affect the reaction barriers. The relevant description can be seen in 2nd paragraph, p.9.

Figure A7. The calculated potential energy profiles of O_2 hydrogenation to H_2O_2 on the a. pure Cu, b. $Au_{0.02}Cu_{0.98}$, c. $Au_{0.5}Cu_{0.5}$, and d. pure Au. Brown, gold, red and white spheres represent Cu, Au, O, and H atoms, respectively.

5. The most stable adsorption configuration is relevant to the subsequent hydrogenation reaction. It is noted that the authors only present one configuration of O_2 adsorption for each substrate. It is suggested that the authors put all the possible adsorption configurations and adsorption energies in the Supporting Information for references.

Response: Thank the reviewer's thoughtful suggestion. We have considered the side-on and end-on configurations of O_2 molecules on all the possible active sites at Cu(100), $Au_{0.02}Cu_{0.98}$ (100), $Au_{0.5}Cu_{0.5}$ (100), and Au(100) surfaces. According to the calculated adsorption energies, all the results show that the side-on configuration is more stable than the end-on configuration for the O_2 molecule. To make the reader clearer, we have added all the adsorption energy of oxygen molecules on these four different metal surfaces, as shown in below **Figs. A8-A9 and Table A6**. These results are also added in the revised Supporting information

(Supplementary Figs.16-17 and Supplementary Table 9). All the adsorption energies of oxygen molecules at each metal surfaces were also corrected by using the solvent effect. The relevant description can be seen in 2nd paragraph, p.9.

Figure A8. Optimized adsorption structures of O₂ molecule by different configurations on the metal surfaces: (a) side-on and (b) end-on at H site on the Cu(100) surface; (c) side-on at H₁ site and (d) side-on at H₂ site on the Au_{0.02}Cu_{0.98}(100) surface; (e) end-on at H₁ site and (f) end-on at H₂ site on the Au_{0.02}Cu_{0.98}(100) surface. Brown, gold, and red spheres represent Cu, Au, and O atoms, respectively.

Figure A9. Optimized adsorption structures of O_2 molecule by different configurations on the metal surfaces: (a) end-on at H_1 site, (b) end-on at T_1 site, (c) side-on at H_1 site, and (d) side-on at H_2 site on the $Au_{0.5}Cu_{0.5}(100)$ surface; (e) end-on at T site and (f) side-on at H site on the $Au(100)$ surface. Brown, gold, and red spheres represent Cu, Au, and O atoms, respectively.

Table A6. The calculated adsorption energy (E_{ads}) of the adsorption of O_2 molecule by side-on and end-on configurations on all the possible adsorption sites at $\text{Cu}(100)$, $\text{Au}_{0.02}\text{Cu}_{0.98}(100)$, $\text{Au}_{0.5}\text{Cu}_{0.5}(100)$, and $\text{Au}(100)$ surfaces. The adsorption energy values outside and inside the parentheses represent with and without considering the solvent effect.

Surface	Configuration	Site	E_{ads} (eV)	Surface	Configuration	Site	E_{ads} (eV)
Cu(100)	Side-on	H	-2.12 (-1.96)	Au(100)	Side-on	H	-0.42 (-0.33)
	End-on	H	-0.86 (-0.62)		End-on	T	-0.21 (-0.17)
Au_{0.02}Cu_{0.98}(100))	Side-on	H₁	-2.00 (-1.84)	Au_{0.5}Cu_{0.5}(100))	Side-on	H₁	-1.16 (-1.03)
	Side-on	H₂	-1.60 (-1.44)		Side-on	H₂	-1.15 (-1.02)
	End-on	H₁	-0.81 (-0.55)		End-on	H₁	-0.24 (-0.04)
	End-on	H₂	-0.62 (-0.37)		End-on	H₂(\rightarrowT₁)^a	-0.28 (-0.10)
	End-on	T₃	n. s.		End-on	T₂	n. s.

^a The left and right parts of the arrow show that the adsorbate is adsorbed on the original active sites before optimization and on the final active sites after optimization, respectively.

n. s.: not stable (the adsorption energy is positive)

6. The authors used VASP calculating reaction energies. We know that VASP cannot calculate charged particle due to the energy divergency. Thus, it is needed to clarify the calculation of the reaction energy for H_2O_2 to $\text{OH}^- + \text{OH}\cdot$. **Response:** Thank the reviewer's thoughtful comment. The OH^- is that the OH group adsorbs on the metal surfaces, while the $\text{OH}\cdot$ radical means the gas-phase OH group in the vacuum (in the revision, we also correct it with solvent effect). The reaction energy for H_2O_2 to $\text{OH}^- + \text{OH}\cdot$ is only the energy difference between the energy of H_2O_2 and the energy summation of one adsorbed OH and one gas-phase OH group. Furthermore, the $\text{OH}^- + \text{OH}\cdot$ can be clarified by the projected density of states (PDOS). As shown in below **Figure A10 (Supplementary Fig. 18)**, the peaks of the adsorbed OH group have no spin-splitting property, while the gas-phase OH group possesses the spin-splitting character. This reveals that the electron configuration of the adsorbed OH group is fully-filled, but the electron configuration of the gas-phase OH group is half-filled. Thus, the adsorbed OH and the gas-phase OH group can be regarded as OH anion and $\text{OH}\cdot$ radical, respectively.

Figure A10. Calculated projected density of states diagrams of (a) $\text{OH}_{(\text{a})}^-$ and (b) $\text{OH}_{(\text{aq})}\cdot$ on the $\text{Au}_{0.02}\text{Cu}_{0.98}(100)$ surface. The dashed line represents the Fermi-level.

The above description has been added in the revised manuscript as follows (1st paragraph, p.10),
“Furthermore, the reaction energies between the H_2O_2 desorption and the production of $\text{OH}\cdot$ radical were also performed. For the production of $\text{OH}\cdot$ radical, we carried out a Fenton-like reaction as follows:

$\text{H}_2\text{O}_{2(a)} \rightarrow \text{OH}_{(a)}^- + \cdot\text{OH}_{(aq)}$; one $\cdot\text{OH}$ radical can form in the solvent via desorption and the other OH group would adsorb on the surface to form OH^- anion. The $\text{OH}^- + \cdot\text{OH}$ can be clarified by the projected density of states (PDOS). As shown in Supplementary Fig. 18, the peaks of the adsorbed OH group have no spin-splitting property, while the gas-phase OH group possesses the spin-splitting character. This reveals that the electron configuration of the adsorbed OH group is fully-filled, but the electron configuration of the gas-phase OH group is half-filled. Thus, the adsorbed OH and the gas-phase OH group can be regarded as OH anion and $\cdot\text{OH}$ radical, respectively.”

7. Due to that the reaction occurs in the solution, the solvation effect should be considered for the calculation.

Response: Thank the reviewer’s thoughtful suggestion. We have considered the solvation effect by using vasp-sol package in our calculation in the revised manuscript. All the adsorption energies, activation energies, and reaction energies in the potential energy profiles were also corrected by using the solvent effect in the revised manuscript.

Please see the computational details (p.16),

“Furthermore, to realistically simulate the effect of the catalytic environment in the solution, we considered the implicit solvent model by using vasp-sol. The implicit solvent effect of water was adopted in this work, where the corresponding dielectric constant of water was 78.3553.”

8. On page 9, the reason for Au carrying negative charge should result from the difference between their electronegativity rather work function.

Response: Thank the reviewer’s thoughtful suggestion. We have corrected this description as per the reviewer’s suggestion in the revised manuscript. The sentence is revised as follows (2nd paragraph, p.10),

“Since the electronegativity of Au is larger than the Cu, the charge of the Au atoms can become more negative in the Au/Cu⁰ nanocubes.”

9. There should be reference comparison for O₂ adsorption on the pure Cu and Au surface to verify the computational settings.

Response: Thank the reviewer’s suggestion. We have added some references regarding O₂ adsorption on the pure Cu and Au surface,

including both experimental and theoretical reports. Experimentally, Cruickshank *et al.* reported that both O atom and OH⁻ could be chemisorbed on the 4-fold hollow site on the Cu(100) surface (Surface Science Letters, 1983, 281, 308-314). Compared with our calculated results, the adsorption of the O₂ molecule on the 4-fold hollow site *via* side-on configuration is also the most stable adsorption structure of O₂ on the Cu(100) surface. Our results also reveal that the most stable adsorption structures of the OH group and O atom are all on the 4-fold hollow site on the Cu(100) surface. Thus, the trends of our computational results are in good agreement with the experimental observations. Theoretically, Gómez *et al.* have calculated that the most stable adsorption geometries of O₂, O, OH, and H₂O species on the Cu(100) surface are the hollow, hollow, hollow, and top sites, respectively (Surface Science, 2021, 714, 121920). These results also have similar trends to our results.

For pure Au surface, Kim and Gewirth have observed the oxygen molecular adsorption on Au(100) surface *via* the SERS study (J. Phys. Chem. B 2006, 110, 2565-2571). In addition, some theoretical studies have reported that the most stable adsorption site on Au(100) surface for oxygen molecule is at the hollow site (J. Phys. Chem. B 2006, 110, 2565-2571; J. Chem. Phys. 2018, 148, 024701). A recent DFT study by Oguz *et al.* showed that the adsorption energy of O₂ on the hollow site at Au(100) surface with PBE functional and DFT-D3 correction is -0.36 eV (J. Chem. Phys. 2018, 148, 024701), which is similar to our results. Hence, according to these comparisons, our computational setting is reliable.

The above description has been added in the revised computational details as references (p.16-17),

“To verify our computational settings, we have compared our results with some references regarding O₂ adsorption on the pure Cu and Au surface, including both experimental and theoretical reports. Experimentally, Cruickshank *et al.* reported that both O atom and OH⁻ could be chemisorbed on the 4-fold hollow site on the Cu(100) surface. Compared with our calculated results, the adsorption of the O₂ molecule on the 4-fold hollow site *via* side-on configuration is also the most stable adsorption structure of O₂ on the Cu(100) surface. Our results also reveal that the most stable adsorption structures of the OH group and O atom are all on the 4-fold hollow site on the Cu(100) surface. Thus, the trends of our computational results are in good agreement with the experimental

observations. Theoretically, Gómez *et al.* have calculated that the most stable adsorption geometries of O₂, O, OH, and H₂O species on the Cu(100) surface are the hollow, hollow, hollow, and top sites, respectively. These results also have similar trends to our results.

For pure Au surface, Kim and Gewirth have observed the oxygen molecular adsorption on Au(100) surface *via* the SERS study. In addition, some theoretical studies have reported that the most stable adsorption site on Au(100) surface for oxygen molecule is at the hollow site. A recent DFT study by Oguz *et al.* showed that the adsorption energy of O₂ on the hollow site at Au(100) surface with PBE functional and DFT-D3 correction is -0.36 eV, which is similar to our results. Hence, according to these comparisons, our computational setting is reliable.”

Reviewers' Comments:

Reviewer #1:

Remarks to the Author:

The authors have responded well to all the concerns and supplemented the relevant experimental evidence, so we suggest that this manuscript can be accepted for publication in Nature Communications.

Reviewer #2:

Remarks to the Author:

The manuscript can be accepted for publication.

Reviewer #1

The authors have responded well to all the concerns and supplemented the relevant experimental evidence, so we suggest that this manuscript can be accepted for publication in Nature Communications.

Response: We thank the reviewer for the positive comments to the manuscript for publication in Nature Communications.

Reviewer #2

The manuscript can be accepted for publication.

Response: We thank the reviewer for the positive comments to the manuscript.